# Geometrically Coupled Monte Carlo Sampling

**Mark Rowland**[*]
University of Cambridge
mr504@cam.ac.uk

**Krzysztof Choromanski**[*]
Google Brain Robotics
kchoro@google.com

**François Chalus**
University of Cambridge
chalusf3@gmail.com

**Aldo Pacchiano**
University of California, Berkeley
pacchiano@berkeley.edu

**Tamás Sarlós**
Google Research
stamas@google.com

**Richard E. Turner**
University of Cambridge
ret26@cam.ac.uk

**Adrian Weller**
University of Cambridge
Alan Turing Institute
aw665@cam.ac.uk

## Abstract

Monte Carlo sampling in high-dimensional, low-sample settings is important in many machine learning tasks. We improve current methods for sampling in Euclidean spaces by avoiding independence, and instead consider ways to couple samples. We show fundamental connections to optimal transport theory, leading to novel sampling algorithms, and providing new theoretical grounding for existing strategies. We compare our new strategies against prior methods for improving sample efficiency, including quasi-Monte Carlo, by studying discrepancy. We explore our findings empirically, and observe benefits of our sampling schemes for reinforcement learning and generative modelling.

## 1 Introduction and related work

Monte Carlo (MC) methods are popular in many areas of machine learning, including approximate Bayesian inference (Robert and Casella, 2005; Rezende et al., 2014; Kingma and Welling, 2014; Welling and Teh, 2011), reinforcement learning (RL) (Salimans et al., 2017; Choromanski et al., 2018c; Mania et al., 2018), and random feature approximations for kernel methods (Rahimi and Recht, 2007; Yu et al., 2016). Typically, Monte Carlo samples are drawn independently. In many applications, however, there may be an imbalance between the computational cost in drawing MC samples from the distribution of interest, and the subsequent cost incurred due to downstream computation with the samples. For example, when a sample represents the configuration of weights in a policy network for an RL problem, the cost of computing forward passes, backpropagating gradients through the network, and interacting with the environment, is much greater than drawing the sample itself. Since a high proportion of total time is spent *computing* with each sample relative to the cost of *generating* the sample, it may be possible to improve efficiency by replacing the default of independent, identically distributed samples by samples with some non-trivial *coupling*.

Such approaches have been studied in computational statistics for decades, often under the guise of variance reduction. Related methods such as control variates, quasi-Monte Carlo (QMC) (Halton, 1960; Aistleitner and Dick, 2015; Dick et al., 2015; Brauchart and Dick, 2012; Sloan and Wozniakowski, 1998; Avron et al., 2016)), herding (Chen et al., 2010; Huszar and Duvenaud, 2012) and

---
[*]Equal contribution

antithetic sampling (Hammersley and Morton, 1956; Salimans et al., 2017) have also been explored. Methods used in recent machine learning applications include orthogonality constraints (Yu et al., 2016; Choromanski et al., 2018b,c, 2017, 2018a). In this paper, we investigate improvements to MC sampling through carefully designed joint distributions, with an emphasis on the low-sample, high-dimensional regime, which is often relevant for practical machine learning applications (Rezende et al., 2014; Kingma and Welling, 2014; Salimans et al., 2017). We call our approach *Geometrically Coupled Monte Carlo* (GCMC) since, as we will see, it is geometrically motivated. Importantly, we focus on Monte Carlo sampling, in contrast to (pseudo-)deterministic approaches such as QMC and herding, as unbiasedness of estimators is often an important property for stochastic approximation. Whilst approaches such as herding and QMC are known to have superior asymptotic performance to Monte Carlo methods in low dimensions, this may not hold in high-dimensional, low-sample regimes, where they do not provide any theoretical improvement guarantees.

We summarize our main contributions below. Throughout the paper, we save proofs of our results for the Appendix; where appropriate, we provide proof sketches to aid intuition.

- We frame the problem of finding an optimal coupling amongst a collection of samples as a multi-marginal transport (MMT) problem: this generalises the notion of optimal transport, which has seen many applications in machine learning (see for example Arjovsky et al., 2017). We show several settings where the MMT problem can be solved analytically. We recover some existing coupling strategies (based on orthogonal matrices), and derive novel strategies, involving coupling norms of pairs of samples.

- To connect to QMC, we show that sets of geometrically coupled Monte Carlo samples give rise to low discrepancy sequences. To our knowledge, we present the first explanation of the success of structured orthogonal matrices for scalable RBF kernel approximation via discrepancy theory.

- We provide exponentially small upper bounds on failure probabilities for estimators of gradients of Gaussian smoothings of blackbox functions based on the gradient sensing mechanism, both for unstructured and orthogonal settings (Choromanski et al., 2018c). These methods can be used to learn good quality policies for reinforcement learning tasks.

- We empirically measure the discrepancy of sequences produced by our method and show that they enable us to learn good quality policies for quadruped robot navigation in low-sample, high-dimensional regimes, where standard QMC approaches based on Halton sequences and related constructions fail.

## 2 Optimal couplings, herding, and optimal transport

Consider the problem of computing the expectation $I_f = \mathbb{E}_{X \sim \eta}[f(X)]$, where $\eta \in \mathscr{P}(\mathbb{R}^d)$ is a multivariate probability distribution and $f : \mathbb{R}^d \to \mathbb{R}$ is some measurable function in $L^1(\eta)$. A standard Monte Carlo approach is to approximate $I_f$ by $\widehat{I}_f^{\text{iid}} = \frac{1}{m} \sum_{i=1}^m f(X_i)$, where the samples $X_1, \ldots, X_m \sim \eta$ are taken independently. This estimator is clearly unbiased. The main question that we are interested in is what joint distributions (or couplings) over the ensemble of samples $(X_1, \ldots, X_m)$ lead to estimators of the expectation above which are still unbiased, but have lower *mean squared error* (MSE) than the i.i.d. estimator $\widehat{I}_f^{\text{iid}}$, defined for a general estimator $\widehat{I}_f$ by:

$$\text{MSE}(\widehat{I}_f) = \mathbb{E}\left[\left(\widehat{I}_f - I_f\right)^2\right]. \tag{1}$$

For sufficiently rich functions classes $\mathscr{F} \subseteq L^2(\eta)$, a coupling of the random variables $(X_1, \ldots, X_m)$ that achieves optimal MSE simultaneously for all functions $f \in \mathscr{F}$ need not exist. We illustrate this with examples in the Appendix Section 8.2. This motivates the approach below to define optimality of a coupling by taking into account average performance across a function class of interest.

### 2.1 *K*-optimal couplings

We begin by defining formally the notion of coupling.

**Definition 2.1.** Given a probability distribution $\eta \in \mathscr{P}(\mathbb{R}^d)$ and $m \in \mathbb{N}$, we denote by $\Lambda_m(\eta)$ the set of all joint distributions of $m$ random variables $(X_1, \ldots, X_m)$, where each random variable $X_i$

has the marginal distribution $\eta$. More formally,

$$\Lambda_m(\eta) = \left\{ \mu \in \mathscr{P}(\mathbb{R}^{d \times m}) \,|\, (\pi_i)_\# \mu = \eta \text{ for } i = 1, \ldots, m \right\},$$

where $\pi_i : \mathbb{R}^{d \times m} \to \mathbb{R}^d$ denotes projection onto the $i^{\text{th}}$ set of $d$ coordinates, for $i = 1, \ldots, m$.

Note that if $X_{1:m} \sim \mu \in \Lambda_m(\eta)$, then because of the restriction on the marginals of $X_{1:m}$, the estimator $m^{-1} \sum_{i=1}^m f(X_i)$ is unbiased for $\mathbb{E}_{X \sim \eta}[f(X)]$, for any $f \in L^1(\eta)$.

We now define the following notion of optimality of a coupling. Similar notions have appeared in the literature when samples are taken to be non-random, or when selecting importance distributions, sometimes referred to as kernel quadrature (Rasmussen and Ghahramani, 2003; Briol et al., 2017).

**Definition 2.2** ($K$-**optimal coupling**). Given a kernel $K : \mathbb{R}^d \times \mathbb{R}^d \to \mathbb{R}$, a $K$-*optimal coupling* is a solution to the optimisation problem

$$\underset{\mu \in \Lambda_m(\eta)}{\arg\min} \; \mathbb{E}_{f \sim \mathrm{GP}(0,K)} \left[ \mathbb{E}_{X_{1:m} \sim \mu} \left[ \left( \frac{1}{m} \sum_{i=1}^m f(X_i) - I_f \right)^2 \right] \right]. \qquad (2)$$

That is, a $K$-optimal coupling is one that gives the best MSE on average when the function concerned is drawn from the Gaussian process $\mathrm{GP}(0, K)$. For background on Gaussian processes, see (Rasmussen and Williams, 2005).

**Remark 2.3.** There are measure-theoretic subtleties in making sure that the objective in Expression (2) is well-defined. For readability, we treat these issues in the Appendix (Section 7), but remark here that it is sufficient to restrict to kernels $K$ for which sample paths of the corresponding Gaussian process are continuous, which we do for the remainder of the paper.

Our ultimate aim is to characterise $K$-optimal couplings under a variety of conditions algorithmically to enable practical implementation. We discuss the identification of $K$-optimal couplings, along with precise statements of algorithms, in Section 2.3. First we develop the theoretical properties of $K$-optimal couplings, starting with the intimate connection between $K$-optimal couplings and multi-marginal transport theory (Pass, 2014). This theory is a generalisation of optimal transport theory to the case where there are more than two marginal distributions.

**Theorem 2.4.** The optimisation problem defining $K$-optimality in Equation (2) is equivalent to the following *multi-marginal transport problem*:

$$\underset{\mu \in \Lambda_m(\eta)}{\arg\min} \; \mathbb{E}_{X_{1:m} \sim \mu} \left[ \sum_{i \neq j} K(X_i, X_j) \right].$$

**Remark 2.5.** The optimal transport problem of Theorem 2.4 has an interesting difference from most optimal transport problems arising in machine learning: in general, its cost function is *repulsive*, so it seeks a transport plan where transport paths are typically *long*, as opposed to the short transport paths sought when the cost is given by e.g. a metric. Intuitively, the optimal transport cost rewards *space-filling* couplings, for which it is uncommon to observe collections of samples close together.

## 2.2 Minimax couplings and herding

Definition 2.2 ($K$-optimality) considers best average-case behaviour. We could instead use a "minimax" definition of optimality, by examining best worst-case behaviour.

**Definition 2.6** (**Minimax coupling**). Given a function class $\mathscr{F} \subseteq L^2(\eta)$, we say that $\mu \in \Lambda_m(\eta)$ is an $\mathscr{F}$-*minimax coupling* if it is a solution to the following optimisation problem:

$$\underset{\mu \in \Lambda_m(\eta)}{\arg\min} \; \underset{f \in \mathscr{F}}{\sup} \; \mathbb{E}_{X_{1:m} \sim \mu} \left[ \left( \frac{1}{m} \sum_{i=1}^m f(X_i) - I_f \right)^2 \right]. \qquad (3)$$

In general, the minimax coupling objective appearing in Equation (3) is intractable. However, there is an elegant connection to concepts from the kernel herding literature that may be established by taking the function class $\mathscr{F}$ to be the unit ball in some reproducing kernel Hilbert space (RKHS).

**Proposition 2.7.** Suppose that the function class $\mathscr{F}$ is the unit ball in some RKHS given by a kernel $K : \mathbb{R}^d \times \mathbb{R}^d \to \mathbb{R}$. Then the component

$$\sup_{f \in \mathscr{F}} \mathbb{E}_{X_{1:m} \sim \mu} \left[ \left( \frac{1}{m} \sum_{i=1}^m f(X_i) - I_f \right)^2 \right]$$

of the minimax coupling objective in Equation (3) may be upper-bounded by the following objective:

$$\mathbb{E}_{X_{1:m} \sim \mu} \left[ \left\| \theta_K \left( \frac{1}{m} \sum_{i=1}^m \delta_{X_i} \right) - \theta_K (\eta) \right\|_{\mathcal{H}_K}^2 \right] , \qquad (4)$$

where $\theta_K : \mathscr{P}(\mathbb{R}^d) \to \mathcal{H}_K$ is the kernel mean embedding into the RKHS $\mathcal{H}_K$ associated with $K$.

We note the intimate connection of the objective in Equation (4) with maximum mean discrepancy (MMD) (Gretton et al., 2012) and herding (Chen et al., 2010; Huszar and Duvenaud, 2012). First, the integrand appearing in Equation (4) is exactly the MMD-squared between $m^{-1} \sum_{i=1}^m \delta_{X_i}$ and $\eta$ with respect to the kernel $K$. Second, if we instead take $m^{-1} \sum_{i=1}^m \delta_{X_i}$ to be a *non-random* measure of the form $m^{-1} \sum_{i=1}^m \delta_{x_i}$, viewing Expression (4) as a function of the delta locations $x_1, \ldots, x_k$ results in exactly the herding optimisation problem. A connection between variance-reduced sampling and herding has also been noted in the context of random permutations (Lomelí et al., 2018). As well as these similarities, there are important differences between herding and the notion described here. Because all samples are regarded as random variables which are constrained to be marginally distributed according to $\eta$, a coupling maintains the usual unbiasedness guarantees of finite-sample Monte Carlo estimators. In contrast, herding is theoretically supported by fast asymptotic rates of convergence for a wide variety of estimators, but because samples are chosen in a deterministic way, estimator properties based on finite numbers of herding samples are harder to describe statistically. Often there are good reasons to eschew unbiasedness of an estimator in favour of fast convergence rates; however, unbiasedness of gradient estimators is crucial in optimisation algorithms performing correctly, as is well-established in the stochastic approximation literature. Bellemare et al. (2017) provide a discussion of this phenomenon in the context of generative modelling.

Interestingly, the following result shows that solutions of Problem (4) coincide exactly with $K$-optimal couplings of Definition 2.2.

**Theorem 2.8.** Given a probability distribution $\eta \in \mathscr{P}(\mathbb{R}^d)$ and a kernel $K : \mathbb{R}^d \times \mathbb{R}^d \to \mathbb{R}$, a coupling $\mu \in \Lambda_m(\eta)$ is $K$-optimal iff it is solves the optimisation problem in Expression (4).

Connections similar to Theorem 2.8 have previously been established in the study of identifying *deterministic* quadrature points (Paskov, 1993) – we also highlight (Kanagawa et al., 2018) as a recent review of such connections. In contrast, here we take random quadrature points with fixed marginal distributions.

## 2.3 Solving for *K*-optimal couplings

In this section, we study the objective defining $K$-optimal couplings, as given in Definition 2.2. The problem is intractable to solve analytically in general, so we present several solutions in settings with additional restrictions, either on the number of samples $m$ in the problem, or on the types of couplings considered. The theoretical statements are given in Theorems 2.9 and 2.10, with the corresponding practical algorithms given as Algorithms 1 and 2. We emphasise that solving Problem (2) in general remains an interesting direction for future work.

**Theorem 2.9.** Let $\eta \in \mathscr{P}(\mathbb{R}^d)$ be isotropic, and let $K : \mathbb{R}^d \times \mathbb{R}^d \to \mathbb{R}$ be a stationary isotropic kernel, such that $K(\mathbf{x}, \mathbf{y})$ is a strictly decreasing, strictly convex function of $\|\mathbf{x} - \mathbf{y}\|$. Then the $K$-optimal coupling of 2 samples $(X_1, X_2)$ from $\eta$ is given by first drawing $X_1 \sim \eta$, and then setting the direction of $X_2$ to be opposite to that of $X_1$, and setting the norm of $\|X_2\|$ so that

$$F_R(\|X_2\|) + F_R(\|X_1\|) = 1 , \qquad (5)$$

where $F_R$ is the CDF associated with the norm of a random vector distributed according to $\eta$.

The proof of this theorem can be found in the Appendix Section 9 and relies on first showing that any optimal coupling must be antithetic and second that an antithetic coupling must satisfy equation

| **Algorithm 1** Antithetic inverse lengths coupling of Theorem 2.9 | **Algorithm 2** Orthogonal coupling of Theorem 2.10 |
|---|---|
| **for** $i = 1, \ldots, m$ **do** <br>   Draw $X_i \sim \eta$. <br>   Set $X_{m+i} = -X_i \frac{F_R^{-1}(1 - F_R(\|X_i\|))}{\|X_i\|}$. <br> **end for** <br> **Output:** $X_1, \ldots, X_{2m}$ marginally $\eta$ distributed, with low MSE. | **for** $i = 1, \ldots, m$ **do** <br>   Draw $X_i \sim \eta$ conditionally orthogonal to $X_1, \ldots, X_{i-1}$. <br>   Set $X_{m+i} = -X_i$. <br> **end for** <br> **Output:** $X_1, \ldots, X_{2m}$ marginally $\eta$ distributed, with low MSE. |

(5) in order for the marginals to be equal to $\eta$. In the Appendix Section 8 we illustrate with a counterexample that the convexity assumption is required. Indeed if most of the mass of $\eta$ is near the origin and the RBF kernel is larger around 0 then the classical antithetic coupling $X_2 = -X_1$ performs better.

Further extending the above situation, we restrict our attention to antithetic couplings and establish that the optimal way to couple $m$ antithetic pairs $(X_i, X_{m+i}) = (X_i, -X_i)$ is to draw sequentially orthogonal samples if the dimension of the space allows it and the marginal $\eta$ is spherically symmetric. Introduce the following notation for the set of antithetic couplings with independent lengths:

$$\Lambda_{2m}^{\text{anti}}(\eta) = \{ \text{Law}(X_1, \ldots, X_{2m}) \in \Lambda_{2m}(\eta) | \ \|X_i\|, 1 \le i \le m \text{ are independent}, X_i = -X_{m+i} \} .$$

**Theorem 2.10.** Let $\eta \in \mathscr{P}(\mathbb{R}^d)$ be isotropic and let $K : \mathbb{R}^d \times \mathbb{R}^d \to \mathbb{R}$ be a stationary isotropic kernel, such that $K(\mathbf{x}, \mathbf{y}) = \Phi(\|\mathbf{x} - \mathbf{y}\|^2)$, where $\Phi$ is a decreasing, convex function. If $\text{Law}(X_1, \ldots, X_{2m})$, with $m \le d$, is a solution to the constrained optimal coupling problem

$$\underset{\mu \in \Lambda_{2m}^{\text{anti}}(\eta)}{\arg\min} \ \mathbb{E}_{X_{1:2m} \sim \mu} \left[ \sum_{i,j=1}^{2m} \Phi \left( \|X_i - X_j\|^2 \right) \right],$$

then it satisfies $\langle X_i, X_j \rangle = 0$ a.s. for all $1 \le i < j \le m$.

The proof of this theorem can be found in the Appendix Section 9 and relies on reformulating the objective function and showing that the exact minimum is attained thanks to convexity. This result illustrates the advantage that orthogonal samples can have over i.i.d. samples, see (Yu et al., 2016) for earlier such settings. Details on how to efficiently sample orthogonal samples can be found in (Stewart, 1980); exact simulation of $d$ orthogonal samples is possible in $\mathcal{O}(d^3)$ time, whilst empirically good quality samples can be obtained from approximate algorithms in $\mathcal{O}(d^2 \log d)$ time. We emphasise that we focus on applications where these increases in sampling costs are insignificant relative to the downstream costs of computing with the samples (such as simulating rollouts in RL environments, as in Section 5.1). However, we note that an interesting direction for future work would be to incorporate a notion of computational complexity into the $K$-optimality objective, to trade off statistical efficiency against sampling costs.

## 3  Low discrepancy of geometrically coupled samples

Having described our notions of optimal couplings in the previous section and obtained several sampling schemes, we now provide an interesting connection between our geometrically coupled samples and low discrepancy sequences that are studied in the QMC literature. Our main interest is in the *local discrepancy function* $\text{disr}_S : \mathbb{R}^d \to \mathbb{R}$ parametrised by a given set of samples $S = \{X_1, ..., X_{|S|}\}$ and defined as follows:

$$\text{disr}_S(\mathbf{u}) = \text{Vol}(J_\mathbf{u}) - \frac{|\{i : X_i \in J_\mathbf{u}\}|}{|S|},$$

where: $J_\mathbf{u} = [0, u_1) \times ... \times [0, u_d)$ and $\text{Vol}(J_\mathbf{u}) = \prod_{j=1}^{d} u_j$. Now define the *star discrepancy function* $D^*(S)$ as: $D^*(S) = \sup_{\mathbf{u} \in [0,1]^d} |\text{disr}_S(\mathbf{u})|$. This function measures the discrepancy between the empirical sample $S$ from the uniform distribution on a hypercube $[0, 1]^d$.

Consider an expression $I_f = \mathbb{E}_{X \sim \lambda}[f(X)]$, where $\lambda \in \mathscr{P}(\mathbb{R}^1)$, and a set of samples $S = \{X_1, ..., X_{|S|}\}$ that is used in a given (Q)MC estimator to approximate $I_f$. The *star discrepancy function* $D_\lambda^*$ *with respect to a distribution* $\lambda$ is defined on $S$ as: $D_\lambda^*(S) \overset{\text{def}}{=} D^*(F_\lambda(S)) = \sup_{u \in [0,1]} |\text{disr}_{F_\lambda(S)}(u)|$, where $F_\lambda(S) = \{F_\lambda(X_i)\}_{i=1,...,|S|}$ and $F_\lambda$ stands for the cdf function for $\lambda$. In other words, to measure the discrepancy between arbitrary distribution $\lambda \in \mathscr{P}(\mathbb{R}^1)$ and a set of samples $S$, the set of samples is transformed to the interval $[0,1]$ via the cdf $F_\lambda$ and the discrepancy between the uniform distribution on $[0,1]$ and the transformed sequence $F_\lambda(S)$ is calculated.

We will focus here on distributions $\lambda \in \mathscr{P}(\mathbb{R}^1)$, which we call *regular distributions*, corresponding to random variables $X$ defined as $X = \mathbf{g}^\top \mathbf{z}$, where $\mathbf{z} \in \mathbb{R}^d$ is a deterministic vector and $\mathbf{g} \in \mathbb{R}^d$ is taken from some isotropic distribution $\tau$ (e.g. multivariate Gaussian distribution). Regular distributions play an important role in machine learning. It is easy to show that the random feature map approximation of radial basis function (RBF) kernels such as Gaussian kernels can be rewritten as $I_f = \mathbb{E}_{X \sim \lambda}[f(X)]$, where $f(x) \overset{\text{def}}{=} \cos(x)$ and $\lambda$ is a regular distribution (Rahimi and Recht, 2007). To sample points from $\lambda$, we will use the standard set $S_{\text{iid}}$ of independent samples as well as the set of orthogonal samples $S_{\text{ort}}$, where marginal distributions of different $\mathbf{g}_i$ are $\lambda$ but different $\mathbf{g}_i$ are conditioned to be exactly orthogonal (see Choromanski et al., 2018b, for explicit constructions). Our main result of this section shows that local discrepancy $\text{disr}_{F_\lambda(S)}(\mathbf{u})$ for a fixed $\mathbf{u} \in [0,1]^d$ is better concentrated around 0 for regular distributions $\lambda$ if orthogonal sets of samples $S$ are used instead of independent samples. Indeed, in both cases one can obtain exponentially small upper bounds on failure probabilities but these are sharper when orthogonal samples are used.

**Theorem 3.1.** [Local discrepancy & regular distributions] Denote by $S_{\text{iid}}$ a set of independent samples, each taken from a regular distribution $\lambda$ and by $S_{\text{ort}}$ the set of orthogonal samples for that distribution. Let $s = |S_{\text{iid}}| = |S_{\text{ort}}|$. Then for any fixed $u \in [0,1]$ and $a \in \mathbb{R}_+$ the following holds: $\mathbb{P}[|\text{disr}_{F_\lambda(S_{\text{iid}})}(u)| > a] \leq 2e^{-\frac{sa^2}{8}} \overset{\text{def}}{=} p_{\text{iid}}(a)$ and for some $p_{\text{ort}}$ satisfying $p_{\text{ort}} < p_{\text{iid}}$ it holds pointwise: $\mathbb{P}[|\text{disr}_{F_\lambda(S_{\text{ort}})}(u)| > a] \leq p_{\text{ort}}(a)$. Also: $Var(\text{disr}_{F_\lambda(S_{\text{ort}})}(u)) < Var(\text{disr}_{F_\lambda(S_{\text{iid}})}(u))$.

Sharper concentration results regarding local discrepancies translate to sharper concentration results for the star discrepancy function $D_\lambda^*$ via the $\epsilon$-net argument and thus also ultimately to sharper results regarding approximation error of MC estimators using regular distributions via the celebrated Koksma-Hlawka Inequality ((Avron et al., 2016); see Theorem 10.4 in the Appendix).

We conclude that orthogonal samples (special instantiations of the GCMC mechanism) lead to strictly better guarantees regarding the approximation error of $I_f$ for functions $f$ with bounded variation and regular distributions $\lambda$ than standard MC mechanisms. This is the case in particular for random feature map based approximators of RBF kernels. The advantages of orthogonal samples in this setting were partially understood before for certain classes of RBF kernels (Choromanski et al., 2018b; Yu et al., 2016), but to the best of our knowledge, general non-asymptotic results and the connection with discrepancy theory were not known.

In Figure 1 we show a kernel density estimate of the distributions of the $D^*$ discrepancies of 50,000 sample sequences $\left( F_{\mathcal{N}(0,1)}^{-1} \left( \frac{\mathbf{g_i}^T \mathbf{z}}{||z||} \right) \right)_{i=1,...,40}$ for a range of coupling algorithms to generate Gaussian samples $\mathbf{g_i}$. We see that using antithetic samples with coupled lengths as in Algorithm 1 leads to a sequence with lower discrepancy on average. We also observe that coupling the samples to be orthogonal reduces the discrepancy. This

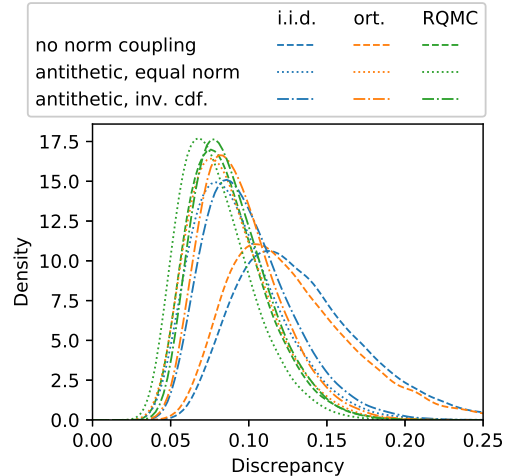

Figure 1: Histograms of the $D^*$ discrepancy for different sampling methods: samples $g_i$ have i.i.d., orthogonal or RQMC directions with uncoupled lengths or lengths coupled according to Algorithms 1 or 2

confirms the above results. Finally this figure
shows that an algorithm designed to have a low
discrepancy (RQMC) will still reach a lower discrepancy than a classical sampling method but this
difference can be mitigated by using antithetic samples.

# 4 Geometric coupling for estimating gradients of function smoothings

Here we provide results on the concentration of zeroth order gradient estimators for reinforcement
learning applications, helping to explain their efficacy. This area is one of the main applications of
the GCMC methods introduced in Section 2, and we present experiments for these applications in
Section 5.1. To our knowledge, we provide the first result showing exponential concentration for the
Evolution Strategies (ES) gradient estimator (Salimans et al., 2017) in this setting. We also provide
exponential concentration bounds for orthogonal gradient estimators.

Recall that given a function $F : \Theta \to \mathbb{R}$ to be minimised, the Vanilla ES gradient estimator is
defined as:

$$\hat{\nabla}_N^V F_\sigma(\theta) = \frac{1}{N\sigma} \sum_{i=1}^{N} F(\theta + \sigma\epsilon_i)\epsilon_i, \quad \text{where } \epsilon_i \sim \mathcal{N}(0, I) \text{ are all i.i.d.} . \tag{6}$$

In what follows we assume that $F$ is uniformly bounded over its domain by $\mathcal{F}$. In the case that $F$ is
a sum of discounted rewards, an upper bound of $\mathcal{R}$ for the reward function yields an upper bound of
$\frac{1}{1-\gamma}\mathcal{R}$ for $F$, where $\gamma$ is the discount factor. Whenever $F$ is bounded in absolute value, the random
vector $\hat{\nabla}_N^V F_\sigma(\theta)$ is sub-Gaussian.

**Theorem 4.1.** If $F$ is a bounded function such that $|F| \leq R_1$, then the vanilla ES estimator is a
sub-Gaussian vector with parameter $\frac{\sqrt{2}R_1\sqrt{8c^2+1}}{\sqrt{N}\sigma}$; with $c = 24e$ and therefore for any $t \geq 0$:

$$\mathbb{P}\left(\max_{j=1,\dots,d} \left|\left(\hat{\nabla}_N^V F_\sigma(\theta)\right)_j - \left(\mathbb{E}\left[\hat{\nabla}_N^V F_\sigma(\theta)\right]\right)_j\right| \geq t\right) \leq 2de^{\frac{-t^2 N\sigma^2}{2R_1^2(8c^2+1)}},$$

for a universal constant $c$.

For the case of pairs of antithetic coupled gradient estimators, one can obtain a similar bound with
comparable performance using this technique.

## 4.1 Bounds for orthogonal estimators

We show that a general class of orthogonal gradient estimators present similar exponential concentra-
tion properties as the Vanilla ES estimator. Proving these bounds is substantially more challenging
because of the correlation structure between samples. To our knowledge, these are the first results
showing exponential concentration for structured gradient estimators, yielding insight as to why
these perform well in practice. We provide concentration bounds for gradient estimators of the
form:

$$\hat{\nabla}_d^{Ort} F(\theta) = \frac{1}{d\sigma} \sum_{i=1}^{d} \nu_i b_i F\left(\theta + \sigma\nu_i b_i\right),$$

where the random vectors $\nu_i \in \mathbb{R}^d$ are sampled uniformly from the unit sphere using a sequentially
orthogonal process, and $b_i$ are zero mean signed lengths, sampled from sub-Gaussian distributions
each with sub-Gaussian parameter $\beta_i$, independent from each other and from all other sources of
randomness. Let $c := 2\sqrt{(24e)^2 + \frac{1}{2}}$. Whenever the function $F$ is bounded, the random variable
vector $\hat{\nabla}_d^{Ort} F(\theta)$ is sub-Gaussian.

**Theorem 4.2.** Let $B = \max_i \mathbb{E}[|b_i|]$, and $\beta = \max_i \beta_i$, $|F| \leq R$, then the orthogonal gradient
estimator $\hat{\nabla}_d^{Ort} F(\theta)$ is sub-Gaussian with parameter $\sqrt{\frac{\beta^2 c^2 R^2}{\sigma^2 d^2} + \frac{R^2 B^2}{4d\sigma^2}}$.

Assuming $N = Td$ and the availability of $T$ i.i.d. orthogonal estimators (indexed by $j$), define:

$$\hat{\nabla}_N^{Ort} F(\theta) = \frac{1}{T} \sum_{j=1}^{T} \hat{\nabla}_d^{Ort,j} F(\theta).$$

**Theorem 4.3.** The gradient estimator $\hat{\nabla}_N^{Ort} F(\theta)$ is sub-Gaussian with parameter $\frac{1}{\sqrt{T}}\sqrt{\frac{\beta^2 c^2 R^2}{\sigma^2 d^2} + \frac{R^2 B^2}{4\sigma^2 d}} = \frac{1}{\sqrt{N}}\sqrt{\frac{\beta^2 c^2 R^2}{d\sigma^2} + \frac{R^2 B^2}{4\sigma^2}}$; and therefore:

$$\mathbb{P}\left(\max_{j=1,\dots,d}\left|\left(\hat{\nabla}_N^{Ort} F(\theta)\right)_j - \left(\mathbb{E}\left[\hat{\nabla}_N^{Ort} F(\theta)\right]\right)_j\right| \geq t\right) \leq 2d e^{\frac{-t^2 N \sigma^2}{\frac{\beta^2 c^2 R^2 \sigma^2}{d} + \frac{R^2 B^2}{4}}}.$$

## 5 Experiments

### 5.1 Learning efficient navigation policies with ES strategies

We consider the task of closed-loop policy optimization to train stable walking behaviors for quadruped locomotion of the Minitaur platform on the Bullet simulator (Coumans and Bai, 2016–2018). We train neural network policies with $d \geq 96$ parameters and optimize the blackbox function $F$ that takes as input parameters of the neural network and outputs the total reward, by applying MC estimators of gradients of Gaussian smoothings of $F$, as described in Expression (6). The main aim of the experiments is to compare policies learnt by using i.i.d. samples, as in Expression (6), against estimators using GCMC methods. We test four different control variate terms that lead to four different variants of the MC algorithm: vanilla (no control variate), forward finite-difference (see Choromanski et al., 2018c, for details), antithetic and antithetic-coupled (see: below). For each of these four variants we use different sampling strategies of calculating the MC estimator: MCGaussian, Halton (baselines), MCGaussianOrthogonal, MCGaussianOrthogonalFixed, and MCRandomHadamard that correspond to: independent Gaussian samples (Salimans et al., 2017), samples constructed from randomized Halton sequences used on a regular basis in QMC methods, Gaussian orthogonal samples (introduced first in Choromanski et al. (2018c) but not tested for $m < d$ and in the locomotion task setting), Gaussian orthogonal samples with renormalized lengths (each length equals $\sqrt{d}$) and finally: rows of random Hadamard matrices (that approximate Gaussian orthogonal samples, but are easier to compute, (see Choromanski et al., 2018c)). For the antithetic variant using Gaussian orthogonal samples, we also test the variant which couples the lengths of antithetic pairs of samples as in Algorithm 1; we refer to this as antithetic $-$ coupled. We tested different number of samples $s$ with the emphasis on MC estimators satisfying: $m \ll d$. We chose: $m = 8, 16, 32, 48, 56, 64, 96$. Full details of the sampling mechanisms described above are given in the Appendix Section.

Figure 2 shows comparison of different MC methods using antithetic variant for $m = 8, 32, 48$ samples given to the MC estimator per iteration of the optimization routine (with an exception of the Halton approach, where we used $m = 96$ samples to demonstrate that even with the larger number of samples standard QMC methods fail). Walkable policies are characterized by total reward $R > 10.0$. We notice that structured approaches outperform the unstructured one and that QMC method based on Halton sequences did not lead to walkable policies. Since it will be also the case for other settings considered by us, we exclude it from the subsequent plots.

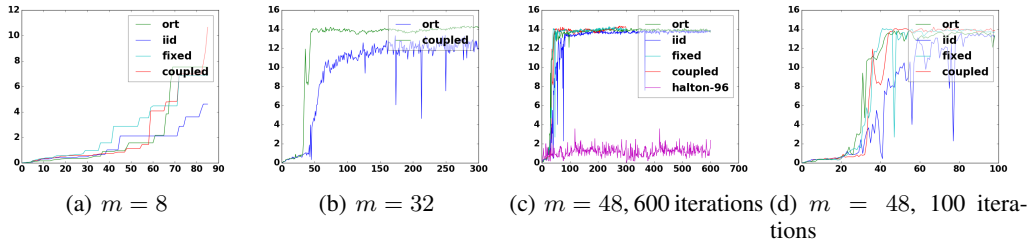

    (a) $m = 8$          (b) $m = 32$      (c) $m = 48$, 600 iterations    (d) $m = 48$, 100 iterations

Figure 2: Training curves for different MC methods. iid, ort, coupled, fixed, halton-96 correspond to: MCGaussian, MCGaussianOrthogonal, antithetic $-$ coupled, MCGaussianOrthogonalFixed and Halton-based QMC method. Subfigure (d) is a zoomed version of Subfigure (c) after just 100 iterations and with Halton approach excluded.

For $m = 32$ we excluded the comparison with MCGaussian since it performed substantially worse than other methods and with MCGaussianOrthogonalFixed since it was very similar to

MCGaussianOrthogonal (for clarity). Again, for clarity, for $m = 8$ we plot the max-reward-curves, where the maximal reward from already constructed policies instead of the current one is plotted (thus these curves are monotonic). In Subfigure (a) the curves stabilize after about $87$ iterations (for the MCGaussianOrthogonal strategy the curve ultimately exceeds reward 10.0 but after $> 500$ iterations).

We conclude that for $m = 8$ the coupling mechanism is the only one that leads to walkable policies and for $m = 32$ it leads to the best policy among all considered structured mechanisms. More experimental results are given in the Appendix. We also attach videos showing how policies learned by applying certain structured mechanisms work in practice (details in the Appendix). Testing all variants of the MC mechanism mentioned above, we managed to successfully train stable walking behaviours using only $m = 8$ samples per iteration only for $k = 5$ settings: MCGaussianOrthogonal-antithetic-coupled, MCGaussianOrthogonal-antithetic, MCGaussianOrthogonal-forward-fd, MCRandomHadamard-antithetic and MCRandomHadamard-vanilla. Thus all 5 policies correspond to some variants of our GCMC mechanism.

We did not conduct hyperparameters tuning to obtain the above curves. We used hyperparameters applied on a regular basis in other Monte Carlo algorithms for policy optimization, in particular chose $\sigma = 0.1$ and $\eta = 0.01$, where $\sigma$ stands for the standard deviation of the entries of Gaussian vectors used for MC and $\eta$ is the gradient step size. The experiments where conducted in a distributed environment on a cluster of machines, where each machine was responsible for evaluating exactly one sample.

## 5.2 Variance-reduced ELBO estimation for deep generative models

In this section, we test GCMC sampling strategies on a deep generative modelling application. We consider a variational autoencoder (VAE) (Rezende et al., 2014; Kingma and Welling, 2014) with latent variable $z$ with prior $p(z)$, observed variable $x$ with trainable generative model $p_\theta(x|z)$, and trainable recognition model $q_\phi(z|x)$. In the standard VAE training algorithm, the evidence lower-bound (ELBO) for a single training point $x$ is:

$$\mathbb{E}_{z \sim q_\phi(\cdot|x)} \left[ \log p_\theta(x, z) - \log q_\phi(z|x) \right] .$$

This objective is then optimised by estimating gradients using a combination of $m \in \mathbb{N}$ i.i.d. Monte Carlo samples together with the reparametrisation trick. We adjust the training algorithm by using a variety of GCMC sampling algorithms, rather than i.i.d. sampling. We train on MNIST, and report the average train and test ELBO after 50 epochs for a variety of sampling algorithms and numbers of samples $K$, to understand the effect of these sampling methods on speeding up learning. The full results and experiment specifications are given in the Appendix Section 12. We observe that GCMC methods consistently lead to better log-likelihoods than i.i.d. sampling, in fact with GCMC methods with 2 samples performing better than i.i.d. methods using 8 samples. We highlight concurrent work (Buchholz et al., 2018) that presents an in-depth study of quasi-Monte Carlo integration for variational inference.

# 6  Conclusion

We have introduced Monte Carlo coupling strategies in Euclidean spaces for improving algorithms that typically operate in a high-dimensional, low-sample regime, demonstrating fundamental connections to multi-marginal transport. In future work, it will be interesting to explore applications in other areas such as random feature kernel approximation. We also highlight more general solution of the $K$-optimality criterion, and incorporation of a sampling cost penalty into the corresponding objective as interesting problems left open by this paper.

# Acknowledgements

We thank Jiri Hron, María Lomelí, and the anonymous reviewers for helpful comments on the manuscript. We thank Yingzhen Li for her VAE implementation. MR acknowledges support by EPSRC grant EP/L016516/1 for the Cambridge Centre for Analysis. AW acknowledges support from the David MacKay Newton research fellowship at Darwin College, The Alan Turing Institute under EPSRC grant EP/N510129/1 & TU/B/000074, and the Leverhulme Trust via the CFI.

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
