[Supplementary Material]

## APPENDIX: Geometrically Coupled Monte Carlo Sampling

## 7  Measure-theoretic considerations regarding *K*-optimality

In this section, we briefly address the measure-theoretic issues arising from the definition of $K$-optimality in Section 2.1, and establish several important integrability results that will be used in the proofs of the results appearing in Section 2. We re-emphasize here that, as stated in Section 2 of the main paper, we restrict to the case where the Gaussian process $\mathrm{GP}(0, K)$ has continuous sample paths in order to avoid unnecessary technical complications. More precisely, we restrict to Gaussian processes for which a continuous modification of the process exists, and assume in the following that it is always this modification that we are working with. The vast majority of commonly used GP kernels in machine learning lead to continuous GP sample paths, with the exception of special cases such as the white noise kernel. For further discussion of the properties of kernels that lead to continuous sample paths in GPs (see e.g. Marcus and Shepp, 1972; Talagrand, 1987; Gin and Nickl, 2015).

We begin by recalling the form of the objective for $K$-optimality, defined in the main paper in Expression (2):

$$\underset{\mu \in \Lambda_m(\eta)}{\arg\min} \ \mathbb{E}_{f \sim \mathrm{GP}(0,K)} \left[ \mathbb{E}_{X_{1:m} \sim \mu} \left[ \left( \frac{1}{m} \sum_{i=1}^m f(X_i) - I_f \right)^2 \right] \right] .$$

Firstly, we establish joint measurability of the random variable $(m^{-1} \sum_{i=1}^m f(X_i) - I_f)^2$. We consider the Gaussian process $f$ taking values in the measurable space $(C(\mathbb{R}^d; \mathbb{R}), \Sigma)$, where $C(\mathbb{R}^d, \mathbb{R})$ is the space of continuous functions from $\mathbb{R}^d$ to $\mathbb{R}$, and $\Sigma$ is the product sigma-algebra, and $(X_1, \ldots, X_m)$ taking values in the measurable space $((\mathbb{R}^d)^m, \mathcal{B}((\mathbb{R}^d)^m))$, where $\mathcal{B}((\mathbb{R}^d)^m)$ is the usual Borel sigma algebra on $(\mathbb{R}^d)^m$. We aim to establish that $(m^{-1} \sum_{i=1}^m f(X_i) - I_f)^2$ is measurable on $(C(\mathbb{R}^d; \mathbb{R}) \times (\mathbb{R}^d)^m, \Sigma \otimes \mathcal{B}((\mathbb{R}^d)^m))$. Considering the types of terms that result from expanding $(m^{-1} \sum_{i=1}^m f(X_i) - I_f)^2$, we note that it is sufficient to prove joint measurability of terms of the form $f(X_i)$ and $f(X_i)f(X_j)$. We deal explicitly with the term $f(X_i)$ here; the treatment of the term $f(X_i)f(X_j)$ is analogous. To do this, we show that the evaluation function $\psi : (C(\mathbb{R}^d; \mathbb{R}) \times (\mathbb{R}^d)^m, \Sigma \otimes \mathcal{B}((\mathbb{R}^d)^m)) \to (\mathbb{R}, \mathcal{B}(\mathbb{R}))$ defined by $\psi(g, x) = g(x)$, is measurable, and since $f(X_i)$ is given by the composition $\psi(f, X_i)$, we have that $f(X_i)$ is measurable, as required. To reach this conclusion, we will show that (i) for all $x \in \mathbb{R}^d$, the function $\psi(\cdot, x) : (C(\mathbb{R}^d; \mathbb{R}), \Sigma) \to \mathbb{R}$ is measurable, and (ii) for all $g \in C(\mathbb{R}^d; \mathbb{R})$, the function $\psi(g, \cdot) : \mathbb{R}^d \to \mathbb{R}$ is continuous. $\psi$ is then said to be a Carathéodory function, and joint measurability follows (see Aliprantis and Border, 2006, Lemma 4.51). For (i), simply note that for all $A \in \mathcal{B}(\mathbb{R})$, $x \in \mathbb{R}^d$, the set $\{g \in C(\mathbb{R}^d; \mathbb{R}) | g(x) \in A\}$ is a cylinder set, and hence in $\Sigma$. In addition, (ii) follows immediately by continuity of $g \in C(\mathbb{R}^d; \mathbb{R})$. Putting all this together, we have established joint measurability of the random variable $(m^{-1} \sum_{i=1}^m f(X_i) - I_f)^2$, which means that the $K$-optimality objective appearing in Definition 2.2 is well-defined.

Finally, we make several remarks on pathwise integrability of the Gaussian process $\mathrm{GP}(0, K)$. Firstly, we show that $f \in L^2(\eta)$ almost surely, via the following calculation:

$$\mathbb{E}_{X \sim \eta} \left[ \mathbb{E}_{f \sim \mathrm{GP}(0,K)} \left[ f(X)^2 \right] \right] = \mathbb{E}_{X \sim \eta} \left[ K(X, X) \right] < \infty,$$

with the bound following from stationarity of the kernel $K$. Thus, we may apply Fubini's theorem to obtain

$$\mathbb{E}_{f \sim \mathrm{GP}(0,K)} \left[ \mathbb{E}_{X \sim \eta} \left[ f(X)^2 \right] \right] = \mathbb{E}_{X \sim \eta} \left[ \mathbb{E}_{f \sim \mathrm{GP}(0,K)} \left[ f(X)^2 \right] \right] .$$

It therefore follows that $f \in L^2(\eta)$ almost surely (under the law of the Gaussian process). The joint integrability with respect to $\eta$ and the law of the Gaussian process established above is important in justifying the use of Fubini's theorem in exchanging orders of expectation in the analysis presented in Section 9.

## 8  Examples and counterexamples relating to Section 2

In this section, we present several examples that serve to further illustrate results from the main text.

## 8.1 Convexity of the kernel is needed in Theorem 2.9

We present a counterexample illustrating that the convexity assumption on the kernel is required in theorem 2.9. Let $K(\mathbf{x}, \mathbf{y}) = e^{-||\mathbf{x}-\mathbf{y}||^2}$ be the Gaussian kernel and let $\eta \in \mathscr{P}(\mathbb{R}^d)$ be the spherically symmetric distribution such that if $X \sim \eta$ then $||X|| \sim \mathcal{U}([0, b])$ for some $b > 0$. Note that the kernel is not convex. Depending on the value of $b$, two different couplings of the norms will be optimal: either $||X_2|| = ||X_1||$ or $||X_2|| = F_R^{-1}(1 - F_R(||X_1||))$. Indeed it is easy to compute numerically the following expectations for this choice of $\eta$ and to note that

$$\mathbb{E}[K(X_1, -X_1)] < \mathbb{E}\left[K\left(X_1, \frac{X_1}{||X_1||} F_R^{-1}(1 - F_R(||X_1||))\right)\right] \iff b < \frac{3}{4}.$$

This illustrates that in the absence of convexity of the kernel the optimal choice of coupling for two samples also depends on $\eta$: if $\eta$ assigns a lot of mass to a small area around 0 ($b$ small) then the coupling of Theorem 2.9 suffers a lot in conjunction with the Gaussian kernel. On the other hand if $\eta$ spreads out mass more evenly further from 0 ($b$ large) then the antithetic coupling giving equal norms to both samples performs better.

## 8.2 Examples illustrating non-existence of uniformly optimal couplings

Below, we give two examples that expand on the remarks in Section 2 stating that in general, there does not exist a coupling of Monte Carlo samples $(X_1, \ldots, X_m)$ that achieves optimal MSE simultaneously for a range of functions $f$ for the objective appearing in Expression (1).

**Example 8.1.** Suppose we want to estimate the value of the following expectation:

$$\mathbb{E}_{X \sim N(0, I)}[f(X)],$$

where $f : \mathbb{R}^2 \to \mathbb{R}$ is specified in polar coordinates by

$$f(r, \theta) = \mathbb{1}_{\theta \in [0, \pi)}.$$

The exact value of this integral is $1/2$. An optimal coupling for two samples marginally distributed as $N(0, I)$ in this case can be shown to be any distribution for which $X_1$ and $X_2$ point in opposite directions almost surely (e.g. taking $X_2 = -X_1$). It is readily checked that the corresponding Monte Carlo estimator is in fact exact, having a mean squared error (MSE) of 0.

**Example 8.2.** Consider the same setup as Example 8.1, but now with the function

$$f(r, \theta) = \mathbb{1}_{\theta \in [0, \pi/2) \cup [\pi, 3\pi/2)}.$$

In this case, it is straightforward to show that the two coupled samples of Example 8.1 obtain the same MSE as a single Gaussian sample, whereas two i.i.d. Gaussian samples obtain half this MSE.

Examples 8.1 and 8.2 illustrate that it is not always possible for one coupling to outperform all others (in terms of MSE) across a given function class.

# 9 Proofs of results in Section 2

**Theorem 2.4.** The optimisation problem defining $K$-optimality in Equation (2) is equivalent to the following *multi-marginal transport problem*:

$$\arg\min_{\mu \in \Lambda_m(\eta)} \mathbb{E}_{X_{1:m} \sim \mu}\left[\sum_{i \neq j} K(X_i, X_j)\right].$$

*Proof.* We calculate as follows, beginning with the $K$-optimality objective:

$$\mathbb{E}_{f\sim\text{GP}(0,K)}\left[\mathbb{E}_{X_{1:m}\sim\mu}\left[\left(\frac{1}{m}\sum_{i=1}^{m}f(X_i)-I_f\right)^2\right]\right]$$

$$=\mathbb{E}_{f\sim\text{GP}(0,K)}\left[\mathbb{E}_{X_{1:m}\sim\mu}\left[\left(\frac{1}{m}\sum_{i=1}^{m}f(X_i)-\mathbb{E}_{X\sim\eta}[f(X)]\right)^2\right]\right]$$

$$=\mathbb{E}_{f\sim\text{GP}(0,K)}\left[\mathbb{E}_{X_{1:m}\sim\mu}\left[\frac{1}{m^2}\sum_{i=1}^{m}f^2(X_i)+\frac{1}{m^2}\sum_{i\neq j}f(X_i)f(X_j)\right.\right.$$

$$\left.\left.-\frac{2}{m}\mathbb{E}_{X\sim\eta}[f(X)]\sum_{i=1}^{m}f(X_i)+\mathbb{E}_{X\sim\eta}[f(X)]^2\right]\right].$$

Removing terms which depend only on the fixed marginal distribution $\eta$, and not the joint distribution $\mu$, the observe that up to a function of $\eta$ only, the objective above is equivalent to

$$\mathbb{E}_{f\sim\text{GP}(0,K)}\left[\mathbb{E}_{X_{1:m}\sim\mu}\left[\sum_{i\neq j}f(X_i)f(X_j)\right]\right].$$

By Fubini's theorem, we obtain

$$\mathbb{E}_{f\sim\text{GP}(0,K)}\left[\mathbb{E}_{X_{1:m}\sim\mu}\left[\sum_{i\neq j}f(X_i)f(X_j)\right]\right]=\mathbb{E}_{X_{1:m}\sim\mu}\left[\mathbb{E}_{f\sim\text{GP}(0,K)}\left[\sum_{i\neq j}f(X_i)f(X_j)\right]\right]$$

$$=\mathbb{E}_{X_{1:m}\sim\mu}\left[\sum_{i\neq j}K(X_i,X_j)\right],$$

as required. $\qquad\square$

**Proposition 2.7.** Suppose that the function class $\mathscr{F}$ is the unit ball in some RKHS given by a kernel $K:\mathbb{R}^d\times\mathbb{R}^d\to\mathbb{R}$. Then the component

$$\sup_{f\in\mathscr{F}}\mathbb{E}_{X_{1:m}\sim\mu}\left[\left(\frac{1}{m}\sum_{i=1}^{m}f(X_i)-I_f\right)^2\right]$$

of the minimax coupling objective in Equation (3) may be upper-bounded by the following objective:

$$\mathbb{E}_{X_{1:m}\sim\mu}\left[\left\|\theta_K\left(\frac{1}{m}\sum_{i=1}^{m}\delta_{X_i}\right)-\theta_K(\eta)\right\|_{\mathcal{H}_K}^2\right],\qquad(4)$$

where $\theta_K:\mathscr{P}(\mathbb{R}^d)\to\mathcal{H}_K$ is the kernel mean embedding into the RKHS $\mathcal{H}_K$ associated with $K$.

*Proof.* We begin by observing that for $f\in\mathcal{H}_K$,

$$\mathbb{E}_{X_{1:m}\sim\mu}\left[\left(\frac{1}{m}\sum_{i=1}^{m}f(X_i)-I_f\right)^2\right]=\mathbb{E}_{X_{1:m}}\left[\left(\int_{\mathbb{R}^d}f(x)\left(\frac{1}{m}\sum_{i=1}^{m}\delta_{X_i}-\eta\right)(\mathrm{d}x)\right)^2\right]$$

$$=\mathbb{E}_{X_{1:m}}\left[\left\langle f,\theta_K\left(\frac{1}{m}\sum_{i=1}^{m}\delta_{X_i}\right)-\theta_K(\eta)\right\rangle_{\mathcal{H}_K}^2\right].$$

Using this observation, we next observe that

$$\sup_{f\in\mathscr{F}}\mathbb{E}_{X_{1:m}\sim\mu}\left[\left(\frac{1}{m}\sum_{i=1}^{m}f(X_i)-I_f\right)^2\right]=\sup_{f\in\mathscr{F}}\mathbb{E}_{X_{1:m}\sim\mu}\left[\left\langle f,\theta_K\left(\frac{1}{m}\sum_{i=1}^{m}\delta_{X_i}\right)-\theta_K\left(\eta\right)\right\rangle_{\mathcal{H}_K}^2\right]$$

$$\leq\mathbb{E}_{X_{1:m}\sim\mu}\left[\sup_{f\in\mathscr{F}}\left\langle f,\theta_K\left(\frac{1}{m}\sum_{i=1}^{m}\delta_{X_i}\right)-\theta_K\left(\eta\right)\right\rangle_{\mathcal{H}_K}^2\right].$$

We can now evaluate the supremum; it is realised when $f\in\mathcal{H}_K$ is the unit vector in the direction of $\theta_K\left(m^{-1}\sum_{i=1}^{m}\delta_{X_i}\right)-\theta_K\left(\eta\right)$, in which case the squared inner product evaluates to $\|\theta_K\left(m^{-1}\sum_{i=1}^{m}\delta_{X_i}\right)-\theta_K\left(\eta\right)\|_{\mathcal{H}_K}^2$. Substituting this in yields the result. $\square$

**Theorem 2.8.** Given a probability distribution $\eta\in\mathscr{P}(\mathbb{R}^d)$ and a kernel $K:\mathbb{R}^d\times\mathbb{R}^d\to\mathbb{R}$, a coupling $\mu\in\Lambda_m(\eta)$ is $K$-optimal iff it is solves the optimisation problem in Expression (4).

*Proof.* The optimisation objective

$$\mathbb{E}_{X_{1:m}\sim\mu}\left[\left\|\theta_K\left(\frac{1}{m}\sum_{i=1}^{m}\delta_{X_i}\right)-\theta_K\left(\eta\right)\right\|_{\mathcal{H}_K}^2\right]$$

can be rewritten in the following form:

$$\mathbb{E}_{X_{1:m}\sim\mu}\left[\left\|\theta_K\left(\frac{1}{m}\sum_{i=1}^{m}\delta_{X_i}\right)-\theta_K\left(\eta\right)\right\|_{\mathcal{H}_K}^2\right]$$

$$=\mathbb{E}_{X_{1:m}\sim\mu}\left[\left\|\frac{1}{m}\sum_{i=1}^{m}K(X_i,\cdot)-\theta_K\left(\eta\right)\right\|_{\mathcal{H}_K}^2\right]$$

$$=\mathbb{E}_{X_{1:m}\sim\mu}\left[\left\|\frac{1}{m}\sum_{i=1}^{m}K(X_i,\cdot)\right\|_{\mathcal{H}_K}^2-2\left\langle\frac{1}{m}\sum_{i=1}^{m}K(X_i,\cdot),\theta_K\left(\eta\right)\right\rangle_{\mathcal{H}_K}+\|\theta_K\left(\eta\right)\|_{\mathcal{H}_K}^2\right].$$

Since the marginal distributions of $X_1,\ldots,X_m$ are fixed, the only term above that depends on the *coupling* between the random variables $X_1,\ldots,X_m$ is the first term. Thus, minimising the original objective is equivalent to minimising

$$\mathbb{E}_{X_{1:m}\sim\mu}\left[\left\|\frac{1}{m}\sum_{i=1}^{m}K(X_i,\cdot)\right\|_{\mathcal{H}_K}^2\right].$$

Expanding this term yields:

$$\mathbb{E}_{X_{1:m}\sim\mu}\left[\frac{1}{m^2}\sum_{i=1}^{m}K(X_i,X_i)+\frac{1}{m^2}\sum_{i\neq j}K(X_i,X_j)\right].$$

Again, the only term depending on the coupling is the final term, so minimising the original objective is equivalent to the following optimisation problem:

$$\min_{\mu\in\Lambda_m(\eta)}\mathbb{E}_{X_{1:m}\sim\mu}\left[\sum_{i\neq j}K(X_i,X_j)\right],$$

over all joint distribution of $X_{1:m}$ with marginals given by $\eta$. $\square$

**Theorem 2.9.** Let $\eta\in\mathscr{P}(\mathbb{R}^d)$ be isotropic, and let $K:\mathbb{R}^d\times\mathbb{R}^d\to\mathbb{R}$ be a stationary isotropic kernel, such that $K(\mathbf{x},\mathbf{y})$ is a strictly decreasing, strictly convex function of $\|\mathbf{x}-\mathbf{y}\|$. Then the $K$-optimal coupling of 2 samples $(X_1,X_2)$ from $\eta$ is given by first drawing $X_1\sim\eta$, and then setting the direction of $X_2$ to be opposite to that of $X_1$, and setting the norm of $\|X_2\|$ so that

$$F_R(\|X_2\|)+F_R(\|X_1\|)=1,\tag{5}$$

where $F_R$ is the CDF associated with the norm of a random vector distributed according to $\eta$.

*Proof.* To demonstrate that the optimal coupling takes the form given in the statement of Theorem, we proceed in two steps: (i) we show that there exists an optimal coupling $\mu^*$ such that if $(X_1, X_2) \sim \mu^*$, then $X_1/\|X_1\| = -X_2/\|X_2\|$ almost surely, i.e. $X_1$ and $X_2$ point in opposite directions almost surely, and (ii) that the norms of $X_1$ and $X_2$ satisfy $\|X_2\| = F_R^{-1}(1 - F_R(\|X_1\|))$ almost surely. We begin with (i).

Let $\mu \in \Lambda_2(\eta)$ be optimal for the following optimisation problem:

$$\min_{\mu \in \Lambda_2(\eta)} \mathbb{E}_{(X_1, X_2) \sim \mu} \left[ K(X_1, X_2) \right] , \tag{7}$$

Then note that if we let $(X_1, X_2) \sim \mu$, then if $R$ is a random matrix draw independently from Haar measure on the orthogonal group $O_d$, then $\mu' = \mathrm{Law}((RX_1, -RX_1/\|X_1\| \times \|X_2\|))$ still lies in $\Lambda_2(\eta)$, and moreover yields an objective value for (7) at least as small as that achieved by $\mu$. The former claim comes from observing that since $\eta$ is radially symmetric, we have $\mathrm{Law}(RX_1) = \mathrm{Law}(X_1) = \eta$, and $\mathrm{Law}(-RX_1 \times \|X_2\|/\|X_1\|)$ is the law of a random vector with uniformly random direction (given by $-RX_1/\|X_1\|$), and independent norm given by $\|X_2\|$, and so is again distributed according to $\eta$. For the latter claim, note that we have

$$\|X_1 - X_2\| \leq \left\| X_1 - \left( -X_1 \frac{\|X_2\|}{\|X_1\|} \right) \right\| = \left\| RX_1 - \left( -RX_1 \frac{\|X_2\|}{\|X_1\|} \right) \right\| ,$$

and so by the assumption of the theorem that $K(\mathbf{x}, \mathbf{y})$ is a decreasing function of $\|\mathbf{x} - \mathbf{y}\|$, we have

$$K(X_1, X_2) \geq K \left( RX_1, -RX_1 \frac{\|X_2\|}{\|X_1\|} \right) ,$$

as required. We have therefore demonstrated that there exists an optimal coupling of $X_1, X_2 \sim \eta$ for (7) such that the vectors $X_1$ and $X_2$ point in opposite directions almost surely.

To establish claim (ii), we note that in order for the coupling to be optimal, under the condition of convexity of $K$, we must have $\|X_2\|$ decreasing monotonically as $\|X_1\|$ increases. It therefore follows that the optimal coupling must of the form stated in the theorem. □

**Theorem 2.10.** Let $\eta \in \mathscr{P}(\mathbb{R}^d)$ be isotropic and let $K : \mathbb{R}^d \times \mathbb{R}^d \to \mathbb{R}$ be a stationary isotropic kernel, such that $K(\mathbf{x}, \mathbf{y}) = \Phi(\|\mathbf{x} - \mathbf{y}\|^2)$, where $\Phi$ is a decreasing, convex function. If $\mathrm{Law}(X_1, \dots, X_{2m})$, with $m \leq d$, is a solution to the constrained optimal coupling problem

$$\operatorname*{arg\,min}_{\mu \in \Lambda_{2m}^{\mathrm{anti}}(\eta)} \mathbb{E}_{X_{1:2m} \sim \mu} \left[ \sum_{i,j=1}^{2m} \Phi \left( \|X_i - X_j\|^2 \right) \right] ,$$

then it satisfies $\langle X_i, X_j \rangle = 0$ a.s. for all $1 \leq i < j \leq m$.

*Proof.* Because $X_i = -X_{i+m}$ a.s., the objective function can be rewritten as follows:

$$\sum_{i,j=1}^{2m} \Phi \left( \|X_i - X_j\|^2 \right) = \sum_{i=1}^{2m} \Phi \left( \|X_i - X_i\|^2 \right) + 2 \sum_{i=1}^{2m-1} \sum_{j=i+1}^{2m} \Phi \left( \|X_i - X_j\|^2 \right)$$

$$= 2m\Phi(0) + 2 \sum_{i=1}^{m-1} \sum_{j=i+1}^{m} \Phi \left( \|X_i - X_j\|^2 \right) + 2 \sum_{i=m+1}^{2m-1} \sum_{j=i+1}^{2m} \Phi \left( \|X_i - X_j\|^2 \right)$$

$$+ 2 \sum_{i=1}^{m} \sum_{j=m+1}^{2m} \Phi \left( \|X_i - X_j\|^2 \right)$$

$$= 2m\Phi(0) + 4 \sum_{i=1}^{m-1} \sum_{j=i+1}^{m} \Phi \left( \|X_i - X_j\|^2 \right) + 2 \sum_{i,j=1}^{m} \Phi \left( \|X_i + X_j\|^2 \right)$$

$$= 2m\Phi(0) + 2 \sum_{i=1}^{m} \Phi \left( \|2X_i\|^2 \right) + 4 \sum_{1 \leq i < j \leq m} \Phi \left( \|X_i - X_j\|^2 \right) + \Phi \left( \|X_i + X_j\|^2 \right) .$$

Hence it becomes equivalent to minimise with $||X_i||$ fixed

$$
E\left[\sum_{1\leq i<j\leq m}\Phi\left(||X_i-X_j||^2\right)+\Phi\left(||X_i+X_j||^2\right)\right]
$$

$$
=E\left[\sum_{1\leq i<j\leq m}\Phi\left(||X_i||^2+||X_j||^2-2\langle X_i,X_j\rangle\right)+\Phi\left(||X_i||^2+||X_j||^2+2\langle X_i,X_j\rangle\right)\right]. \quad (8)
$$

Using convexity note that

$$
2\Phi\left(||\mathbf{x}||^2+||\mathbf{y}||^2\right)=2\Phi\left(\frac{||\mathbf{x}||^2+||\mathbf{y}||^2-2\langle\mathbf{x},\mathbf{y}\rangle+||\mathbf{x}||^2+||\mathbf{y}||^2+2\langle\mathbf{x},\mathbf{y}\rangle}{2}\right)
$$

$$
\leq\Phi\left(||\mathbf{x}||^2+||\mathbf{y}||^2-2\langle\mathbf{x},\mathbf{y}\rangle\right)+\Phi\left(||\mathbf{x}||^2+||\mathbf{y}||^2+2\langle\mathbf{x},\mathbf{y}\rangle\right)
$$

and equality is attained whenever $\langle\mathbf{x},\mathbf{y}\rangle=0$. Therefore, the expectation in equation (8) is minimized when $\langle X_i,X_j\rangle=0$ a.s. for $1\leq i<j\leq m$. This is a set of valid constraints as long as $d\geq m$. Also, if $X_1\sim\eta$ and the $X_j$'s are generated under $\eta$ conditioned on being orthogonal to $X_1,\ldots,X_{j-1}$ then $X_j$ also has marginal $\eta$ because drawing a uniform distribution conditional on being orthogonal to a uniform axis results in a uniform direction. Hence any optimal coupling must satisfy this condition. $\qquad\square$

## 10 Proofs of results in Section 3

For the reader's convenience we restate our main result that we prove here.

**Theorem 3.1.** [Local discrepancy & regular distributions] Denote by $S_{\text{iid}}$ a set of independent samples, each taken from a regular distribution $\lambda$ and by $S_{\text{ort}}$ the set of orthogonal samples for that distribution. Let $s=|S_{\text{iid}}|=|S_{\text{ort}}|$. Then for any fixed $u\in[0,1]$ and $a\in\mathbb{R}_+$ the following holds: $\mathbb{P}[|\text{disr}_{F_\lambda(S_{\text{iid}})}(u)|>a]\leq 2e^{-\frac{sa^2}{8}}\overset{\text{def}}{=}p_{\text{iid}}(a)$ and for some $p_{\text{ort}}$ satisfying $p_{\text{ort}}<p_{\text{iid}}$ it holds pointwise: $\mathbb{P}[|\text{disr}_{F_\lambda(S_{\text{ort}})}(u)|>a]\leq p_{\text{ort}}(a)$. Also: $Var(\text{disr}_{F_\lambda(S_{\text{ort}})}(u))<Var(\text{disr}_{F_\lambda(S_{\text{iid}})}(u))$.

*Proof.* Consider a set of samples $S=\{X_1,...,X_{|S|}\}$ with elements of marginal distributions $\text{Unif}[0,1]^d$. Denote $s=|S|$. Note that for any given $\mathbf{u}=(u_1,\ldots,u_d)\in[0,1]^d$ we have:

$$
\mathbb{E}[\text{disr}_S(\mathbf{u})]=\prod_{j=1}^d u_j-\frac{\mathbb{E}[|\{i:X_i\in J_\mathbf{u}\}|]}{s}=\prod_{j=1}^d u_j-\frac{\sum_{i=1,\ldots,s}\mathbb{P}[X_i\in J_\mathbf{u}]}{s}
$$

$$
=\prod_{j=1}^d u_j-\frac{s\cdot\text{Vol}(J_\mathbf{u})}{s}=0
$$

Thus we conclude that the expected value of $\text{disr}_S(\mathbf{u})$ is 0. Let $\eta\in\mathscr{P}(\mathbb{R}^d)$ be some isotropic distribution and denote by $\lambda$ the distribution corresponding to the random variable $X=\mathbf{z}^\top\mathbf{g}$, where $\mathbf{z}\in\mathbb{R}^d$ is some fixed deterministic vector and $\mathbf{g}$ is sampled from $\eta$.

Now consider a set of samples $S_{\text{iid}}$ of the form: $S_{\text{iid}}=\{\mathbf{z}^\top\mathbf{g}_1^{\text{iid}},...,\mathbf{z}^\top\mathbf{g}_s^{\text{iid}}\}$, where random vectors $\mathbf{g}_1^{\text{iid}},...,\mathbf{g}_s^{\text{iid}}$ are chosen independently from $\eta$. Similarly, consider a set of samples $S_{\text{ort}}$ of the form: $S_{\text{ort}}=\{\mathbf{z}^\top\mathbf{g}_1^{\text{ort}},...,\mathbf{z}^\top\mathbf{g}_s^{\text{ort}}\}$, where the marginal distributions of the random vectors $\mathbf{g}_1^{\text{ort}},...,\mathbf{g}_s^{\text{ort}}$ is $\eta$, but this time different $\mathbf{g}_i^{\text{ort}}$ are conditioned to be orthogonal.

For any given $u\in[0,1]$ we have the following:

$$
\text{disr}_{\psi(S_{\text{iid}})}(u)=u-\frac{\sum_{i\in S}I[\psi(\mathbf{z}^\top\mathbf{g}_i^{\text{iid}})<u]}{s},
$$

where $\psi\overset{\text{def}}{=}F_\lambda$ and $I$ is an indicator random variable.

Note that equivalently, we can rewrite $\mathrm{disr}_{\psi(S_{\mathrm{iid}})}(u)$ as:

$$\mathrm{disr}_{\psi(S_{\mathrm{iid}})}(u) = u - \frac{-\sum_{i \in S} I[\psi(\mathbf{z}^\top \mathbf{g}_i^{\mathrm{iid}}) \geq u] + s}{s} = u - 1 + \frac{-\sum_{i \in S} I[\psi(\mathbf{z}^\top \mathbf{g}_i^{\mathrm{iid}}) \geq u]}{s}.$$

From our previous analysis and standard properties of the cdf we conclude that $\frac{\sum_{i \in S} I[\psi(\mathbf{z}^\top \mathbf{g}_i^{\mathrm{iid}}) \geq u]}{s}$ is an unbiased estimator of $u - 1$.

Similarly, we obtain:

$$\mathrm{disr}_{\psi(S_{\mathrm{ort}})}(u) = u - 1 + \frac{-\sum_{i \in S} I[\psi(\mathbf{z}^\top \mathbf{g}_i^{\mathrm{ort}}) \geq u]}{s}.$$

Again, as before we note that $\frac{\sum_{i \in S} I[\psi(\mathbf{z}^\top \mathbf{g}_i^{\mathrm{ort}}) \geq u]}{s}$ is an unbiased estimator of $u - 1$ (since the marginal distributions of the $g_i^{\mathrm{ort}}$ are the same as those of the $g_i^{\mathrm{iid}}$). Denote $Y_i^{\mathrm{iid}} = I[\psi(\mathbf{z}^\top \mathbf{g}_i^{\mathrm{iid}}) \geq u]$ and $Y_i^{\mathrm{ort}} = I[\psi(\mathbf{z}^\top \mathbf{g}_i^{\mathrm{ort}}) \geq u]$. Note that $Y_i^{\mathrm{iid}}, Y_i^{\mathrm{ort}} \in \{0, 1\}$ for $i = 1, \ldots, s$. Notice also that even though the random variables $(Y_i^{\mathrm{iid}})_{i=1}^s$ are independent, this is not true of $(Y_i^{\mathrm{ort}})_{i=1}^s$.

We will use the following Cramer's Theorem:

**Theorem 10.1.** Let $Y_1, \ldots, Y_s$ be random variables. Denote: $W_s = \frac{Y_1 + \ldots + Y_s}{s}$. Then $\mathbb{P}[W_s \geq a] \leq \min_{\theta > 0} \frac{\mathbb{E}[e^{s\theta W_s}]}{e^{s\theta a}}$.

Denote: $Z_i^{\mathrm{iid}} = 1 - Y_i^{\mathrm{iid}}$ and similarly: $Z_i^{\mathrm{ort}} = 1 - Y_i^{\mathrm{ort}}$. Denote: $W_s^{\mathrm{iid}} = \frac{Y_1^{\mathrm{iid}} + \ldots + Y_s^{\mathrm{iid}}}{s}$, $W_s^{\mathrm{iid},-} = \frac{Z_1^{\mathrm{iid}} + \ldots + Z_s^{\mathrm{iid}}}{s}$ and similarly: $W_s^{\mathrm{ort}} = \frac{Y_1^{\mathrm{ort}} + \ldots + Y_s^{\mathrm{ort}}}{s}$, $W_s^{\mathrm{ort},-} = \frac{Z_1^{\mathrm{ort}} + \ldots + Z_s^{\mathrm{ort}}}{s}$. Note that for any $0 < c < 1$:

$$\mathbb{P}[|W_s^{\mathrm{iid}} - \mathbb{E}[W_s^{\mathrm{iid}}]| > c] = \mathbb{P}[W_s^{\mathrm{iid}} > \mathbb{E}[W_s^{\mathrm{iid}}] + c] + \mathbb{P}[W_s^{\mathrm{iid}} < \mathbb{E}[W_s^{\mathrm{iid}}] - c].$$

Denote: $\mu = \mathbb{E}[W_s^{\mathrm{iid}}]$. We get:

$$\mathbb{P}[|W_s^{\mathrm{iid}} - \mathbb{E}[W_s^{\mathrm{iid}}]| > c] = \mathbb{P}[W_s^{\mathrm{iid}} > \mu + c] + \mathbb{P}[W_s^{\mathrm{iid},-} > \mu_- + c],$$

where $\mu_- = 1 - \mu = \mathbb{E}[W_s^{\mathrm{iid},-}]$.

Therefore, using Cramer's Theorem, we get:

$$\mathbb{P}[|W_s^{\mathrm{iid}} - \mathbb{E}[W_s^{\mathrm{iid}}]| > c] \leq \min_{\theta > 0} \frac{\mathbb{E}[e^{s\theta W_s^{\mathrm{iid}}}]}{e^{s\theta a_1}} + \min_{\theta > 0} \frac{\mathbb{E}[e^{s\theta W_s^{\mathrm{iid},-}}]}{e^{s\theta a_2}},$$

where: $a_1 = \mu + c$ and $a_2 = \mu_- + c$. Thus, from independence and the fact that all $Y_j^{\mathrm{iid}}$ have the same distribution, we have:

$$\mathbb{P}[|W_s^{\mathrm{iid}} - \mathbb{E}[W_s^{\mathrm{iid}}]| > c] \leq \min_{\theta > 0} \frac{(\mathbb{E}[e^{\theta Y_1^{\mathrm{iid}}}])^s}{e^{s\theta a_1}} + \min_{\theta > 0} \frac{(\mathbb{E}[e^{\theta Z_1^{\mathrm{iid}}}])^s}{e^{s\theta a_2}}. \tag{9}$$

Notice that: $|\mathrm{disr}_{\psi(S_{\mathrm{iid}})}(u)| = |W_s^{\mathrm{iid}} - \mathbb{E}[W_s^{\mathrm{iid}}]|$. Straightforward calculation of the RHS of Equation (9) leads to the upper bound on $|\mathrm{disr}_{\psi(S_{\mathrm{iid}})}(u)|$ from Theorem 3.1.

Now notice, that by the same analysis as before, we get:

$$\mathbb{P}[|W_s^{\mathrm{ort}} - \mathbb{E}[W_s^{\mathrm{ort}}]| > c] \leq \min_{\theta > 0} \frac{\mathbb{E}[e^{s\theta W_s^{\mathrm{ort}}}]}{e^{s\theta a_1}} + \min_{\theta > 0} \frac{\mathbb{E}[e^{s\theta W_s^{\mathrm{ort},-}}]}{e^{s\theta a_2}}.$$

Thus to complete the proof of Theorem 3.1, it suffices to show that: $\mathbb{E}[e^{s\theta W_s^{\mathrm{ort}}}] < \mathbb{E}[e^{s\theta W_s^{\mathrm{iid}}}]$ and: $\mathbb{E}[e^{s\theta W_s^{\mathrm{ort},-}}] < \mathbb{E}[e^{s\theta W_s^{\mathrm{iid},-}}]$ for any $\theta > 0$. We will show the first inequality. The proof of the second inequality is completely analogous.

**Lemma 10.2.** The following holds for any fixed $\theta > 0$:

$$\mathbb{E}[e^{s\theta W_s^{\mathrm{ort}}}] < \mathbb{E}[e^{s\theta W_s^{\mathrm{iid}}}].$$

*Proof.* Notice first that:

$$\mathbb{E}[e^{s\theta W_s^{\mathrm{ort}}}] = \mathbb{E}[e^{\theta \sum_{i=1}^{s} Y_i^{\mathrm{ort}}}] = \sum_{j=0}^{\infty} \frac{\theta^j}{j!} \sum_{j_1+...+j_k=j} \sum_{i_1<...<i_k} \mathbb{E}[(Y_{i_1}^{\mathrm{ort}})^{j_1}...(Y_{i_k}^{\mathrm{ort}})^{j_k}]$$

$$= \sum_{j=0}^{\infty} \frac{\theta^j}{j!} \sum_{j_1+...+j_k=j} \sum_{i_1<...<i_k} \mathbb{E}[Y_{i_1}^{\mathrm{ort}}...Y_{i_k}^{\mathrm{ort}}],$$

where the sum over $j_1,...j_k$ is the sum over all partitioning of $j$ into positive integers $j_1,...,j_k$, the sum over $i_1 < ... < i_k$ is the sum over all increasing nonempty sequences $(i_1,...,i_k)$ such that $i_1,...i_k \in \{1,...,s\}$ (thus $k \in \{1,...,s\}$) and furthermore the last equality is true since each $Y_i^{\mathrm{ort}}$ is an indicator random variable (note that the infinite sum above is well-defined since random variables under consideration are indicators). Similarly,

$$\mathbb{E}[e^{s\theta W_s^{\mathrm{iid}}}] = \mathbb{E}[e^{\theta \sum_{j=1}^{s} Y_j^{\mathrm{iid}}}] = \sum_{j=0}^{\infty} \frac{\theta^j}{j!} \sum_{j_1+...+j_k=j} \sum_{i_1,...,i_k} \mathbb{E}[Y_{i_1}^{\mathrm{iid}}...Y_{i_k}^{\mathrm{iid}}],$$

Thus, since $\theta > 0$, it suffices to show that:

$$\mathbb{E}[Y_{i_1}^{\mathrm{ort}}...Y_{i_k}^{\mathrm{ort}}] < \mathbb{E}[Y_{i_1}^{\mathrm{iid}}...Y_{i_k}^{\mathrm{iid}}]$$

or equivalently:

$$\mathbb{P}[\mathcal{A}_{i_1}^{\mathrm{ort}} \wedge ... \wedge \mathcal{A}_{i_k}^{\mathrm{ort}}] < \mathbb{P}[\mathcal{A}_{i_1}^{\mathrm{iid}} \wedge ... \wedge \mathcal{A}_{i_k}^{\mathrm{iid}}],$$

where $\mathcal{A}_i^{\mathrm{iid}}$ and $\mathcal{A}_i^{\mathrm{ort}}$ stand for events corresponding to indicators $Y_i^{\mathrm{iid}}$ and $Y_i^{\mathrm{ort}}$ respectively. For clarity, we present the proof for the inequality above for $k = 2$, for $k > 2$ the analysis is analogous. Notice also that the inequality for $k = 2$ immediately leads to the inequality regarding the variance from the statement of the theorem.

Note that it suffices to show that for any $t$ and any $\mathbf{z} \in \mathbb{R}^d$ the following is true:

$$\mathbb{P}[\mathbf{z}^\top \mathbf{g}_i^{\mathrm{ort}} \geq t \wedge \mathbf{z}^\top \mathbf{g}_j^{\mathrm{ort}} \geq t] < \mathbb{P}[\mathbf{z}^\top \mathbf{g}_i^{\mathrm{iid}} \geq t \wedge \mathbf{z}^\top \mathbf{g}_j^{\mathrm{iid}} \geq t]$$

for $i \neq j$.

Since $t$ is arbitrary, we can assume without loss of generality that $\|\mathbf{z}\| = 1$. Note that then the following holds:

$$\mathbb{P}[\mathbf{z}^\top \mathbf{g}_i^{\mathrm{ort}} \geq t \wedge \mathbf{z}^\top \mathbf{g}_j^{\mathrm{ort}} \geq t] = \mathbb{P}\left[\frac{g_1}{\sqrt{g_1^2 + ... + g_d^2}} l_1 \geq t \wedge \frac{g_2}{\sqrt{g_1^2 + ... + g_d^2}} l_2 \geq t\right],$$

where $\mathbf{g} = (g_1,...,g_n)^\top$ is a multivariate Gaussian vector taken from a distribution $\mathcal{N}(0, I_d)$ and $l_1, l_2$ are taken independently from the distribution of the length of $\mathbf{g}_i^{\mathrm{ort}}$ (or $\mathbf{g}_i^{\mathrm{iid}}$ since the marginal distributions of samples are the same). The last equality immediately follows from the fact that $\mathbf{g}_i^{\mathrm{ort}}$ are taken from the isotropic distribution.

We also have:

$$\mathbb{P}[\mathbf{z}^\top \mathbf{g}_i^{\mathrm{iid}} \geq t \wedge \mathbf{z}^\top \mathbf{g}_j^{\mathrm{iid}} \geq t] = \mathbb{P}[\mathbf{z}^\top \mathbf{g}_i^{\mathrm{iid}} \geq t]\mathbb{P}[\mathbf{z}^\top \mathbf{g}_j^{\mathrm{iid}} \geq t]$$

$$= \mathbb{P}\left[\frac{g_1}{\sqrt{g_1^2 + ... + g_d^2}} l_1 \geq t\right] \cdot \mathbb{P}\left[\frac{g_2}{\sqrt{g_1^2 + ... + g_d^2}} l_2 \geq t\right],$$

where we use the independence assumption. Thus it suffices to show that:

$$\mathbb{P}\left[g_1 \geq \frac{t}{l_1}\sqrt{g_1^2 + ... + g_d^2} \wedge g_2 \geq \frac{t}{l_2}\sqrt{g_1^2 + ... + g_d^2}\right]$$

$$< \mathbb{P}\left[g_1 \geq \frac{t}{l_1}\sqrt{g_1^2 + ... + g_d^2}\right] \cdot \mathbb{P}\left[g_2 \geq \frac{t}{l_2}\sqrt{g_1^2 + ... + g_d^2}\right].$$

Therefore we want to prove that:

$$\mathbb{P}\left[g_2 \geq \frac{t}{l_2}\sqrt{g_1^2 + ... + g_d^2}\middle| g_1 \geq \frac{t}{l_1}\sqrt{g_1^2 + ... + g_d^2}\right] < \mathbb{P}\left[g_2 \geq \frac{t}{l_2}\sqrt{g_1^2 + ... + g_d^2}\right].$$

Note that: $\frac{t}{l_1}, \frac{t}{l_2}, g_1, g_2$ and $g_3^2 + ... + g_d^2$ are independent random variables. Thus it suffices to show that for any $c > 0$ and any $a, b$ the following is true:

$$\mathbb{P}\left[g_2 \geq b\sqrt{g_1^2 + g_2^2 + c}\middle| g_1 \geq a\sqrt{g_1^2 + g_2^2 + c}\right] < \mathbb{P}\left[g_2 > b\sqrt{g_1^2 + g_2^2 + c}\right].$$

Denote $g_i^+ = g_i|g_i > 0$ and $g_i^- = g_i|g_i < 0$. We will prove the inequality above by conditioning on four possible events" $\mathcal{E}_1 = \{g_1, g_2 > 0\}$, $\mathcal{E}_2 = \{g_1, g_2 < 0\}$, $\mathcal{E}_3 = \{g_1 > 0, g_2 < 0\}$, $\mathcal{E}_4 = \{g_1 < 0, g_2 > 0\}$. Consider all four cases, one can easily see that in order to prove the inequality it suffices to prove the following: for any $x, y$:

$$\mathbb{P}[(g_2^+)^2 > x|(g_2^+)^2 \leq y] < \mathbb{P}[(g_2^+)^2 > x],$$

and similarly:

$$\mathbb{P}[(g_2^+)^2 < x|(g_2^+)^2 \geq y] < \mathbb{P}[(g_2^+)^2 < x].$$

We will prove the first inequality. The proof for the second one is completely analogous.

Denote: $A_1 = \mathbb{P}[(g_2^+)^2 < x]$, $A_2 = \mathbb{P}[x < (g_2^+)^2 < y]$ and $A_3 = \mathbb{P}[(g_2^+)^2 > y]$. We want to prove that: $\frac{A_2}{A_1+A_2} < A_2 + A_3$, which is trivially true since $0 < A_1, A_2, A_3 < 1$ and $A_1 + A_2 + A_3 = 1$. That completes the proof of Lemma 10.2. $\qquad \square$

As we have noticed, the proof of Lemma 10.2 completes the proof of Theorem 3.1. $\qquad \square$

## 10.1 From low discrepancy to low approximation error

As mentioned in the main body of the paper, sharper concentration results regarding local discrepancies translate to sharper concentration results for the star discrepancy function $D_\eta^*$ and ultimately also to sharper results regarding approximation error of MC estimators. We show it in this section.

Define the error coming from the approximation $\hat{I}_f$ of $I_f$ that uses the set of samples $S$ as: $\epsilon_S(f) = |I_f - \hat{I}_f|$.

The following theorem establishes the connection between the discrepancy of a sequence $S$ used for estimation and the above approximation error $\epsilon_S(f)$.

**Theorem 10.3** (Koksma-Hlawka inequality)**.** For any function f with bounded variation and a sequence $S$, the approximation error $\epsilon_S(f)$ satisfies:

$$\epsilon_S(f) \leq D_\lambda^*(S)V_{\text{HK}}(f) = \sup_{\mathbf{u} \in [0,1]^d} |\text{disr}_{\psi(S)}(\mathbf{u})|V_{\text{HK}}(f),$$

where $V_{\text{HK}}$ stands for the Hardy-Krause variation of $f$ (Niederreiter, 1992) defined as follows:

$$V_{\text{HK}}(f) = \sum_{I \subset [d], I \neq \emptyset} \int_{[0,1]^d} \left|\frac{\partial f}{\partial \mathbf{u}_I}|_{u_j=1, j \notin I}\right| d\mathbf{u}_I.$$

and $\psi = F_\lambda$.

Thus we can conclude that sequences $S$ of lower discrepancies lead to tighter upper bounds on the approximation error of $I_f$.

The following is our main result of this section:

**Theorem 10.4.** For any $N \in \mathbb{N}, a > \frac{1}{N}$, set of samples $S$, $\lambda \in \mathscr{P}([0,1])$ and a function $f$ of bounded variation the following holds for $\epsilon_S(f) = |\hat{I}_f - I_f|$, where $I_f = \mathbb{E}_{X \sim \eta}[f(X)]$:

$$\mathbb{P}[\epsilon_S(f) > a] \leq Np\left(a - \frac{1}{N}\right)V_{\text{HK}}(f),$$

if $p$ is such that: $\mathbb{P}[|\mathrm{disr}_{\psi(S))}(x)| > a] \le p(a)$ for any fixed $x \in [0, 1]$. In particular, if $\lambda$ is a regular distribution and if we take: $a = \frac{\log(|S|)}{|S|}$ and $N = \frac{2\sqrt{|S|}}{\log(|S|)}$, then we obtain: $\mathbb{P}[\epsilon_{S_{\mathrm{iid}}}(f) > a] \le \frac{4\sqrt{|S|}}{\log(|S|)}e^{-\frac{\log^2(|S|)}{32}} = \mathrm{neg}(|S|)$, and for $S_{\mathrm{iid}}$ replaced by $S_{\mathrm{ort}}$ the bounds are even tighter. In the above statement $\mathrm{neg}(|S|)$ is defined as $\mathrm{neg}(|S|) = \frac{1}{\mathrm{superpoly}(|S|)}$ and superpoly stands for some superpolynomial function.

*Proof.* Let $s = |S|$. Consider $D_\lambda^*(S) = \sup_{x \in [0,1]} |\mathrm{disr}_{\psi(S)}(x)|$ for some one-dimensional distribution $\lambda$. We partition interval $[0, 1]$ into $N$ subintervals: $[x_j, x_{j+1}]$ of length $\frac{1}{N}$ each for $j = 0, ..., N - 1$. Note that for a fixed $a > 0$:

$$\left\{\sup_{x \in [0,1]} |\mathrm{disr}_{\psi(S)}(x)| > a\right\} = \left\{\exists x^* \in [0, 1] : |\mathrm{disr}_{\psi(S)}(x^*)| > a\right\}.$$

Denote: $X_j^x = I[s_j < x]$, where $s_j$ is the $j^{th}$ sample from $\psi(S)$. Assume that $x^*$ is in the subinterval with endpoints: $x_{j^*}$ and $x_{j^*+1}$. Note that we have:

$$\frac{X_1^{x_{j^*}} + ... + X_s^{x_{j^*}}}{s} \le \frac{X_1^{x^*} + ... + X_s^{x^*}}{s} \le \frac{X_1^{x_{j^*+1}} + ... + X_s^{x_{j^*+1}}}{s}$$

Thus we get:

$$\mathrm{disr}_{\psi(S)}(x^*) = \left|\frac{X_1^{x^*} + ... + X_s^{x^*}}{s} - x^*\right| \le \max(A, B),$$

where $A = \left|\frac{X_1^{x_{j^*}} + ... + X_s^{x_{j^*}}}{s} - x^*\right|$ and $B = \left|\frac{X_1^{x_{j^*+1}} + ... + X_s^{x_{j^*+1}}}{s} - x^*\right|$.

Therefore, using triangle inequality, we obtain:

$$\{|\mathrm{disr}_{\psi(S)}(x^*)| > a\} \subseteq \{|\mathrm{disr}_{\psi(S)}(x_{j^*})| + |x^* - x_{j^*}| > a\} \cup \{|\mathrm{disr}_{\psi(S)}(x_{j^*+1})| + |x^* - x_{j^*+1}| > a\}.$$

Therefore we obtain:

$$\left\{\sup_{x \in [0,1]} |\mathrm{disr}_{\psi(S)}(x)| > a\right\} \subseteq \left\{\exists_j : |\mathrm{disr}_{\psi(S)}(x_j)| > a - \frac{1}{N}\right\}.$$

Thus, by the union bound, we conclude that:

$$\mathbb{P}\left[\left\{\sup_{x \in [0,1]} |\mathrm{disr}_{\psi(S)}(x)| > a\right\}\right] \le \sum_{j=1}^N \mathbb{P}\left[\left\{|\mathrm{disr}_{\psi(S)}(x^j)| > a - \frac{1}{N}\right\}\right].$$

The statement of Theorem 10.4 follows now from Koksma-Hlawka inequality, and Theorem 3.1.

$\square$

# 11  Proofs of results in Section 4

## 11.1  Exponential concentration

In this section we present the proofs of the results in 4. For completeness we start by reviewing the classical defnitions of sub-Gaussianity for random variables and random vectors.

**Definition 11.1** (Sub-Gaussian Random Variables)**.** A random variable $X$ with mean $\mu = \mathbb{E}[X]$ is sub-Gaussian if there is a positive number $\sigma$ such that:

$$\mathbb{E}[e^{\lambda(X-\mu)}] \le e^{\sigma^2\lambda^2/2} \text{ for all } \lambda \in \mathbb{R}.$$

A standard Gaussian random variable is sub-Gaussian with parameter $\sigma$ equal to the said variable's standard deviation.

**Lemma 11.2** (Concentration for sub-Gaussian random variables). Let $X$ be a sub-Gaussian random variable with parameter $\sigma$ and mean $\mu$. It satisfies the following concentration inequality:

$$\mathbb{P}[X - \mu \geq t] \leq e^{-\frac{t^2}{2\sigma^2}}.$$

Paying a factor of 2 we can get an equivalent two sided bound for $|X - \mu| \geq t$.

The following alternative characterization of sub-Gaussianity will prove useful:

**Lemma 11.3** (Alternative characterization of sub-Gaussianity ). A *centered* random variable $X$ is sub-Gaussian if there is a constant $c$ and a Gaussian random variable $Z \sim N(0, \tau^2)$ such that:

$$\mathbb{P}\left[|X| \geq s\right] \leq c\mathbb{P}\left[|Z| \geq s\right], \text{ for all } s \geq 0.$$

Additionally, we can switch from the definition in 11.1 to the characterization in Lemma 11.3 in the following way:

- If $X$ is zero mean sub-Gaussian with parameter $\sigma$, then taking $\tau^2 = 2\sigma^2$ and $c = \sqrt{8e}$ is enough for Lemma 11.3 to hold for $X$.

- If $X$ is zero mean sub-Gaussian with sub-Gaussian parameters $\tau^2$ and $c$ in Lemma 11.3, then $\sigma^2 = 2c^2\tau^2$ is a valid sub-Gaussian parameter for for $X$, as in Definition 11.1.

The concept of sub-Gaussianity extends to vector valued random variables:

**Definition 11.4** (Sub-Gaussian Vector). A random vector $X$ is sub-Gaussian with parameter at most $\sigma$ if for every $v \in \mathbb{S}^{d-1}$ (where $\mathbb{S}^{d-1}$ is the unit $d-$dimensional sphere.)

$$\mathbb{E}\left[e^{\lambda\langle v, X\rangle}\right] \leq e^{\frac{\lambda^2\sigma^2}{2}} \text{ for all } \lambda \in \mathbb{R}.$$

We will be using these facts heavily in the following sections.

The following facts will prove useful. For a detailed survey of these results consult (Boucheron et al., 2013).

- Fact 1 If $X$ is sub-Gaussian with parameter $\sigma$, $X + c$ is sub-Gaussian with parameter $\sigma$ for all $c \in \mathbb{R}$.
- Fact 2 If $X_1$ and $X_2$ are independent with parameters $\sigma_1, \sigma_2$ respectively, then $X_1 + X_2$ is sub-Gaussian with parameter $\sqrt{\sigma_1^2 + \sigma_2^2}$.
- Fact 3 Even without assuming independence, if $X_1$ and $X_2$ are sub-Gaussian with parameters $\sigma_1, \sigma_2$ respectively, then $X_1 + X_2$ is sub-Gaussian with parameter $\sqrt{2}\sqrt{\sigma_1^2 + \sigma_2^2}$.
- Fact 4 If $X$ is sub-Gaussian with parameter $\sigma^2$, $cX$ is sub-Gaussian with parameter $c^2\sigma^2$.

We need the following result showing that the product of a sub-Gaussian random variable and a bounded random variable is again sub-Gaussian.

**Theorem 11.5** (Sub-Gaussian products). Let $Y$ be a bounded random variable such that $Y \in [-R_1, R_2]$ for $R_1, R_2 \geq 0$ with $R_1 + R_2 = R$ for some constant $R$, and let $X$ be sub-Gaussian with parameter $\sigma$ and mean $\mu$. Then $XY$ is sub-Gaussian with parameter $\sqrt{2}\sqrt{2g(\sigma)^2 R^2 + \mu^2 R^2/2 + \sigma^2 R_1^2}$ where $g(\sigma) = 24e\sigma$.

Theorem 11.5 allows us to prove fast concentration rates for the vanilla ES estimator. We believe this result is of independent interest, as it tackles a fundamental question regarding concentration of products of sub-Gaussian variables.

*Proof.* The proof has two steps, we first show it holds for non-negative and discrete $Y$. Then we generalize.

1 Case 1: $Y$ only attains discrete values $Y \in \{0, 1\}$ and $X$ is zero mean.

We will make use of lemma 11.3 to prove this result. Since $X$ is mean zero and sub-Gaussian there is $\tau^2$ and constant $c$ such that:

$$P(|X| \geq s) \leq cP(|Z| \geq s),$$

where $Z \sim N(0, \tau^2)$. In fact we can take $\tau^2 = 2\sigma^2$, and $c = \sqrt{8}e$.

Let $X'$ be an independent copy of $X$. By Fact 2, $X - X'$ is sub-Gaussian with parameter $\sqrt{2}\sigma$. Since $X - X'$ has mean zero by Lemma 11.3 we conclude there are constants $\tau_1$ such that $\tau_1^2 = 4\sigma^2$ and $c_1 = \sqrt{8}e$ such that $P(|X - X'| \geq s) \leq c_1 P(|Z'| \geq s)$, where $Z' \sim N(0, \tau_1^2)$.

Let $\mu_{XY}$ denote the mean of $XY$. Let $X'$ and $Y'$ be independent copies of $X$ and $Y'$ respectively. We proceed to invoke a symmetrization argument. We first show that in order to bound the MGF of $XY$ it is enough to bound the MGF of $XY - X'Y'$. For any $\lambda \in \mathbb{R}$:

$$\mathbb{E}\left[e^{\lambda(XY - \mu_{XY})}\right] = \mathbb{E}\left[e^{\lambda(XY - \mathbb{E}[X'Y'])}\right]$$
$$\leq \mathbb{E}\left[e^{\lambda(XY - X'Y')}\right].$$

The inequality follows from Jensen's inequality. This means that sub-Gaussianity of $XY - X'Y'$ implies sub-Gaussianity of $XY - \mu_{XY}$.

We will show sub-Gaussianity of $XY - X'Y'$. Since $Y$ and $Y'$ only take values in $\{0, 1\}$ we can write:

$$|XY - X'Y'| = \begin{cases} |X - X'| & \text{if } Y = 1, Y' = 1, \\ |X| & \text{if } Y = 1, Y' = 0, \\ |X'| & \text{if } Y = 0, Y' = 1, \\ 0 & \text{o.w.} \end{cases}$$

Let $s > 0$. By the union bound:

$$\mathbb{P}(|XY - X'Y'| \geq s) \leq \mathbb{P}(|X - X'| \geq s) + \mathbb{P}(|X| \geq s) + \mathbb{P}(|X'| \geq s).$$

By sub-Gaussianity of $X$ and $X' - X$ and using their Gaussian tail bounds:

$$\mathbb{P}(|XY - X'Y'| \geq s) \leq c_1\mathbb{P}(|Z'| \geq s) + 2c\mathbb{P}(|Z| \geq s),$$

where $Z \sim N(0, \tau^2)$ and $Z' \sim N(0, \tau_1^2)$.

Let $\tau_2 = \max(\tau, \tau_1)$ and $c_2 = 3\max(c, c_1, 1)$. Let $c_2 = 3\sqrt{8}e$ and $\tau_2^2 = 4\sigma^2$. We conclude that for all $s \geq 0$:

$$\mathbb{P}(|XY - X'Y'| \geq s) \leq c_2\mathbb{P}(|Z''| \geq s),$$

where $Z'' \sim N(0, \tau_2^2)$. The inequality also holds for $s = 0$ since we have ensured $c_2 \geq 1$. By the series of observations right below Lemma 11.3, this implies that $XY$ is sub-Gaussian with parameter $\sqrt{2}c_3\tau_2 = \sqrt{2} * 3 * \sqrt{8} * e * 2 * \sigma = 24e\sigma \stackrel{\text{def}}{=} g(\sigma)$.

## 1 Case 2: $X$ centered, $Y \geq 0$ and supported on a finite set $a_1 < a_2 < \cdots < a_m$.

Denote by $\mu_{XY}$ the mean of $XY$. In order to show $XY$ is sub-Gaussian we have to bound its MGF $\mathbb{E}\left[e^{\lambda(XY - \mu_{XY})}\right]$. For $i \geq 1$ let $X_i = X\mathbf{1}(Y \leq a_i)$. Define $a_0 = 0$ then $XY = \sum_{i=1}^{n} X_i(a_i - a_{i-1})$

and therefore $\mathbb{E}[XY] = \sum_{i=1}^{n}(a_i - a_{i-1})\mathbb{E}[X_i]$. Notice that $\sum_{i=1}^{n} a_i - a_{i-1} = a_n - a_0$. Let $p_i = \frac{a_n - a_0}{a_i - a_{i-1}}$. Let $\sigma$ be the sub-Gaussianity parameter of $X$.

$$
\begin{aligned}
\mathbb{E}\left[e^{\lambda(XY - \mu_{XY})}\right] &= \mathbb{E}\left[e^{\sum_{i=1}^{n}\lambda(a_i - a_{i-1})(X_i - \mathbb{E}[X_i])}\right] \\
&\leq \prod_i \mathbb{E}\left[\left(e^{\lambda(a_i - a_{i-1})(X_i - \mathbb{E}[X_i])}\right)^{\frac{a_n - a_0}{a_i - a_{i-1}}}\right]^{\frac{a_i - a_{i-1}}{a_n - a_0}} \\
&= \prod_i \mathbb{E}\left[e^{\lambda(a_n - a_0)(X_i - \mathbb{E}[X_i])}\right]^{\frac{a_i - a_{i-1}}{a_n - a_0}} \\
&\leq \prod_i \left(e^{\frac{\lambda^2(a_n - a_0)^2 g(\sigma)^2}{2}}\right)^{\frac{a_i - a_{i-1}}{a_n - a_0}} \\
&= e^{\frac{\lambda^2 g(\sigma)^2 (a_n - a_0)^2}{2}},
\end{aligned}
$$

where $g(\sigma)$ is defined as in Case 1. The first inequality follows by Hölder's inequality with parameters $p_i$. The second inequality follows from the sub-Gaussianity bound from Case 1 since $X_i = X\mathbb{1}(Y \leq a_i)$.

## 2 Case 2. $Y$ is non-negative but not necessarily discrete. $X$ has mean zero.

Assume $Y \in [0, R]$. If $Y \geq 0$ there is a sequence of simple random variables (all of which are discrete) $Y_n$ with $Y_n \to Y$ almost surely. Furthermore, all $Y_n \in [0, R]$, so that the maximal element in the domain of $Y_n$ is at most $R$.

For any $\lambda \in \mathbb{R}$ the previous observation implies $\lambda X Y_n \to \lambda X Y$ almost surely. Let $f_1(x, y) = |\lambda R X|$. Since $|X|$ is integrable, $\mathbb{E}[f_1(X)] < \infty$. Notice that pointwise $|\lambda X Y_n| \leq f_1(X, Y)$. By the dominated convergence theorem (Halmos, 2013) we can conclude $\mathbb{E}[\lambda X Y_n] \xrightarrow{n \to \infty} \mathbb{E}[\lambda X Y]$. By continuity of the exponential function $h(x) = e^x$ this also implies $e^{\mathbb{E}[\lambda X Y_n]} \xrightarrow{n \to \infty} e^{\mathbb{E}[\lambda X Y]}$.

The random variables $e^{\lambda X Y_n}$ converge to $e^{\lambda X Y}$ almost surely. The function $f_2(x, y) = e^{f_1(x, y)}$ satisfies:

1. $|e^{\lambda X Y_n}| = e^{\lambda X Y_n} \leq f_2(X, Y)$ pointwise.

2. $\mathbb{E}[f_2(X, Y)] < \infty$. Indeed: $\mathbb{E}[f_2(X, Y)] \leq \mathbb{E}[e^{\lambda R X}] + \mathbb{E}[e^{-\lambda R X}] \leq 2e^{\lambda^2 R^2 \sigma^2/2} < \infty$. The first inequality holds by nonnegativity of the exponential function and because for any point $x$, one of $\lambda R x$ or $-\lambda R x$ equals $|\lambda R x|$. The second inequality holds by sub-Gaussianity of $X$.

By the dominated convergence theorem again we conclude that $\mathbb{E}[e^{\lambda X Y_n}] \to \mathbb{E}[e^{\lambda X Y}]$. Since by Case 2, $\mathbb{E}[e^{\lambda X Y_n}] \leq e^{\lambda^2 g(\sigma)^2 R^2/2}$ we conclude $\mathbb{E}[e^{\lambda X Y_n}] \leq e^{\lambda^2 g(\sigma)^2 R^2/2}$.

### Case 3 $X$ has mean $\mu$ and $Y \in [-R_1, R_2]$ can attain negative values.

Define $R := R_1 + R_2$ and let $-R_1$ be the smallest element in the support of $Y$. Let $Y_1 = Y + R_1$ and $X_1 = X - \mu$. Notice $Y_1 \in [0, R]$ and $X_1$ has mean zero and sub-Gaussianity parameter $\sigma$, like $X$. By Case 3 we conclude $X_1 Y_1$ is sub-Gaussian with parameter $g(\sigma)R$.

Since $Y$ is bounded, $Y$ is sub-Gaussian with parameter $R/2$, (see Boucheron et al., 2013). Therefore $\mu Y$ is sub-Gaussian with parameter $|\mu|R/2$.

Since $X$ is sub-Gaussian with parameter $\sigma$, $R_1 X$ is sub-Gaussian with parameter $\sigma R_1$.

Notice that $X_1 Y_1 = XY + R_1 X - \mu Y - R_1 \mu$. Therefore:

1. $X_1 Y_1 + R_1 \mu$ is sub-Gaussian with parameter $g(\sigma)R$ since it is the translate of a $g(\sigma)R$ sub-Gaussian random variable.

2. $X_1 Y_1 + R_1 \mu + \mu Y$ is sub-Gaussian with parameter $\sqrt{2}\sqrt{g(\sigma)^2 R^2 + \mu^2 R^2/4}$ by Fact 3.

3. $X_1 Y_1 + R_1 \mu + \mu Y - R_1 X = XY$ is sub-Gaussian with parameter

$$\sqrt{2}\sqrt{2(g(\sigma)^2 R^2 + \mu^2 R^2/4) + \sigma^2 R_1^2} = \sqrt{2}\sqrt{2g(\sigma)^2 R^2 + \mu^2 R^2/2 + \sigma^2 R_1^2}.$$

This shows that $XY$ is sub-Gaussian with parameter $\sqrt{2}\sqrt{2g(\sigma)^2 R^2 + \mu^2 R^2/2 + \sigma^2 R_1^2}$ which concludes the proof.

$\square$

## 11.2 The vanilla ES estimator

In this section we focus on proving Theorem 4.1. Recall that given $F : \Theta \to \mathbb{R}$, the Vanilla ES estimator is defined as:

$$\hat{\nabla}_N^V F_\sigma(\theta) = \frac{1}{N\sigma} \sum_{i=1}^N F(\theta + \sigma \epsilon_i) \epsilon_i,$$

where $\epsilon_i \sim \mathcal{N}(0, I)$ are all i.i.d. and $\sigma$ is the variance of the length of the sensing direction.

In what follows we assume that $F$ is uniformly bounded over its domain by $\mathcal{F}$. If $F$ is a sum of discounted rewards, an upper bound of $\mathcal{R}$ for the reward function yields an upper bound of $\frac{1}{1-\gamma}\mathcal{R}$ for $F$, where $\gamma$ is the discount factor.

We show that whenever $F$ is bounded in absolute value the random vector $\hat{\nabla}_N^V F_\sigma(\theta)$ is sub-Gaussian.

**Theorem 11.6.** If $F$ is a bounded function such that $|F| \leq R_1$, the vanilla ES estimator is a sub-Gaussian vector with parameter $\frac{\sqrt{2}R_1\sqrt{8c^2+1}}{\sqrt{N}\sigma}$ for $c = 24e$.

*Proof.* Let $v \in \mathbb{S}^{d-1}$ be an arbitrary $d-$dimensional unit vector. We start by showing sub-Gaussianity of the vector $F(\theta + \sigma \epsilon)\epsilon$.

Notice that $\langle v, F(\theta + \sigma \epsilon)\epsilon \rangle = F(\theta + \sigma \epsilon)\langle \epsilon, v \rangle$. Since linear combinations of jointly Gaussian random variables are Gaussian, and $\|v\| = 1$, the random variable $\langle \epsilon, v \rangle$ is a $\mathcal{N}(0, 1)$ Gaussian random variable and therefore $\langle \epsilon, v \rangle$ is 1-sub-Gaussian.

By Theorem 11.5, since $F$ is assumed to be in the range $[-R_1, R_1]$, it follows that $\langle v, F(\theta + \epsilon)\epsilon \rangle$ is $\sqrt{2}R_1\sqrt{8c^2+1}$ sub-Gaussian, where $c = g(1)$.

By noting that each $\epsilon_i$ is independent from all others, we can obtain that $\langle v, \hat{\nabla}_N^V F_\sigma(\theta) \rangle$ is $\frac{\sqrt{2}R_1\sqrt{8g(1)^2+1}}{\sqrt{N}\sigma}$ sub-Gaussian. Since $v$ was arbitrary this concludes the proof.

$\square$

**Corollary 11.7** (Exponential Concentration for the Vanilla ES estimator). If $F$ is a bounded function such that $|F| \leq R_1$:

$$\mathbb{P}\left(\max_{j=1,\cdots,d}\left|\left(\hat{\nabla}_N^V F_\sigma(\theta)\right)_j - \left(\mathbb{E}\left[\hat{\nabla}_N^V F_\sigma(\theta)\right]\right)_j\right| \geq t\right) \leq 2de^{\frac{-t^2 N\sigma^2}{2R_1^2(8g(1)^2+1)}}$$

For any $t \geq 0$.

The combination of Theorem 11.6 and Corollary 11.7 conclude the proof of Theorem 4.1.

## 11.3 Orthogonal Bounds

In this section we prove Theorem 4.3. We prove concentration bounds for orthogonal gradient estimators of the form:

$$\hat{\nabla}_d^{Ort} F(\theta) = \frac{1}{d\sigma} \sum_{i=1}^{d} \nu_i b_i F(\theta + \sigma \nu_i b_i),$$

where the random vectors $\nu_i \in \mathbb{S}^{d-1}$ are in the unit sphere and are sampled uniformly from the unit sphere using a sequentially orthogonal process, the function $F$ is bounded $\sup_x |F(x)| \leq R < \infty$, and $b_i$ are zero mean signed lengths, sampled from sub-Gaussian distributions each with parameter $\beta_i$ and independent from each other and from all other sources of randomness.

**Theorem 11.8.** Let $B = \max_i \mathbb{E}[|b_i|]$, $\beta = \max_i \beta_i$, $|F| \leq R$, the orthogonal gradient estimator $\hat{\nabla}_d^{Ort} F(\theta)$ is sub-Gaussian with parameter $\sqrt{\frac{\beta^2 c^2 R^2}{\sigma^2 d^2} + \frac{R^2 B^2}{4\sigma^2 d}}$. Where $c = 2\sqrt{(24e)^2 + \frac{1}{2}}$, and $\ln(e) = 1$.

*Proof.* We start by lumping in $\frac{1}{\sigma}$ with $F$ so that $|F|/\sigma \leq R/\sigma$. We proceed with the proof, and at the end subsitute $R$ by $R/\sigma$.

In order to show the concentration of the random vector $\hat{\nabla}_d^{Ort} F(\theta)$, it is enough to show that for any fixed $u \in \mathbb{S}^{d-1}$, $\langle u, \hat{\nabla}_d^{Ort} F(\theta) \rangle$ is a sub-Gaussian random variable.

Given $u \in \mathbb{S}^{d-1}$, define $\alpha_i = \langle u, \nu_i \rangle$. Notice that $\sum_{i=1}^{d} \alpha_i^2 = 1$ and that:

$$\langle u, d\hat{\nabla}_d^{Ort} F(\theta) \rangle = \sum_{i=1}^{d} \alpha_i b_i F(\theta + \sigma^2 \nu_i b_i)$$

We wish to control:

$$\mathbb{E}\left[ \exp\left( \lambda \left( \langle u, d\hat{\nabla}_d^{Ort} F(\theta) \rangle - \mathbb{E}\left[ \langle u, d\hat{\nabla}_d^{Ort} F(\theta) \rangle \right] \right) \right) \right].$$

We start by decomposing the MGF above as follows:

$$\mathbb{E}\left[ \exp\left( \lambda \left( \langle u, d\hat{\nabla}_d^{Ort} F(\theta) \rangle - \mathbb{E}\left[ \langle u, d\hat{\nabla}_d^{Ort} F(\theta) \rangle \Big| \nu_1, \cdots, \nu_d \right] + \right.\right.\right.$$
$$\left.\left.\left. \mathbb{E}\left[ \langle u, d\hat{\nabla}_d^{Ort} F(\theta) \rangle \Big| \nu_1, \cdots, \nu_d \right] - \mathbb{E}\left[ \langle u, d\hat{\nabla}_d^{Ort} F(\theta) \rangle \right] \right) \right) \right].$$

And first bounding the conditional MGF:

$$\mathbb{E}\left[ \exp\left( \lambda \left( \langle u, d\hat{\nabla}_d^{Ort} F(\theta) \rangle - \mathbb{E}\left[ \langle u, d\hat{\nabla}_d^{Ort} F(\theta) \rangle \Big| \nu_1, \cdots, \nu_d \right] \right) \right) \Big| \nu_1, \cdots, \nu_d \right].$$

Notice that conditional on $\nu_1, \cdots, \nu_d$, the sum $\langle u, \hat{\nabla}_d^{Ort} F(\theta) \rangle = \sum_{i=1} \alpha_i b_i F(\theta + \sigma^2 \nu_i b_i)$ is made of $d$ (conditionally) independent random variables $\{\alpha_i b_i F(\theta + \sigma^2 \nu_i b_i)\}_{i=1}^{d}$, and therefore, by Theorem 11.5 and Fact 2, the conditional MGF is bounded by:

$$e^{\frac{\lambda^2}{2} \sum_{i=1}^{d} \alpha_i^2 \beta_i^2 c^2 R^2}.$$

For $c = \max_i \frac{\sqrt{2}\sqrt{2g(\beta_i)^2 + \mathbb{E}[b_i]^2/2 + \beta_i^2}}{\beta_i}$ derived from applying Theorem 11.5 to this case, where $\beta_i$ are the sub-Gaussian parameters of the random variables $b_i$. Since $\sum_{i=1}^d \alpha_i^2 = 1$ the bound reduces to:

$$\mathbb{E}\left[\exp\left(\lambda\left(\langle u, d\hat{\nabla}_d^{Ort}F(\theta)\rangle - \mathbb{E}\left[\langle u, d\hat{\nabla}_d^{Ort}F(\theta)\rangle\Big|\nu_1, \cdots, \nu_d\right]\right)\right)\Big|\nu_1, \cdots, \nu_d\right] \leq e^{\frac{\lambda^2}{2}\beta^2 c^2 R^2},$$

where $\beta$ is an uppper bound to $\beta_i$ for all $i$. Notice that this bound has no dependence on dimension. To provide a bound for the MGF:

$$\mathbb{E}\left[\exp\left(\lambda\left(\mathbb{E}\left[\langle u, d\hat{\nabla}_d^{Ort}F(\theta)\rangle\Big|\nu_1, \cdots, \nu_d\right] - \mathbb{E}\left[\langle u, d\hat{\nabla}_d^{Ort}F(\theta)\rangle\right]\right)\right)\right].$$

The random variable:

$$\mathbb{E}\left[\langle u, d\hat{\nabla}_d^{Ort}F(\theta)\rangle\Big|\nu_1, \cdots, \nu_d\right] = \sum_{i=1}^d \mathbb{E}\left[\alpha_i b_i f(\theta + \sigma\nu_i b_i)|\nu_1, \cdots, \nu_d\right]$$

$$= \sum_{i=1}^d \alpha_i \mathbb{E}\left[b_i f(\theta + \sigma\nu_i b_i)|\nu_1, \cdots, \nu_d\right]$$

is bounded. Indeed by Hölder:

$$\left|\sum_{i=1}^d \alpha_i \mathbb{E}\left[b_i f(\theta + \sigma\nu_i b_i)|\nu_1, \cdots, \nu_d\right]\right| \leq RB\|\alpha\|_1,$$

where $\mathbb{E}\left[|b_i|\right] \leq B, \forall i$ and $\alpha \in \mathbb{R}^d$ with $\alpha \in \mathbb{S}^{d-1}$. The later implies $\|\alpha\|_1 \leq \sqrt{d}$. And $R$ is a uniform upper bound for $f$.

The previous observations imply in turn that this random variable is bounded by $RB\sqrt{d}$, and therefore that it is sub-Gaussian because it is bounded (Boucheron et al., 2013). Therefore:

$$\mathbb{E}\left[\exp\left(\lambda\left(\mathbb{E}\left[\langle u, d\hat{\nabla}_d^{Ort}F(\theta)\rangle|\nu_1, \cdots, \nu_d\right] - \mathbb{E}\left[\langle u, d\hat{\nabla}_d^{Ort}F(\theta)\rangle\right]\right)\right)\right] \leq \exp\left(\frac{\lambda^2 R^2 B^2 d}{8}\right).$$

Plugging these two bounds together:
$$\mathbb{E}\left[\exp\left(\lambda\left(\langle u, d\hat{\nabla}_d^{Ort}F(\theta)\rangle - \mathbb{E}\left[\langle u, d\hat{\nabla}_d^{Ort}F(\theta)\rangle\right]\right)\right)\right] \leq e^{\frac{4\lambda^2\beta^2 c^2 R^2 + \lambda^2 R^2 B^2 d}{8}}.$$

As a consequence:

$$\mathbb{E}\left[\exp\left(\lambda\left(\langle u, \hat{\nabla}_d^{Ort}F(\theta)\rangle - \mathbb{E}\left[\langle u, \hat{\nabla}_d^{Ort}F(\theta)\rangle\right]\right)\right)\right] \leq e^{\frac{\lambda^2 \beta^2 c^2 R^2}{2d^2} + \frac{\lambda^2 R^2 B^2}{8d}}.$$

With $c = \max_i \sqrt{2}\sqrt{2 * g(\beta_i)^2 + \mathbb{E}[b_i]^2/2 + \beta_i^2}$ with $g$ the function defined in Theorem 11.5.

$\square$

Assuming $N = Td$ and therefore availability of $T$ i.i.d. orthogonal estimators (indexed by $j$) define:

$$\hat{\nabla}_N^{Ort}F(\theta) = \frac{1}{T}\sum_{j=1}^T \hat{\nabla}_d^{Ort,j}F(\theta).$$

The following corollary is immediate:

**Corollary 11.9.** The gradient estimator $\hat{\nabla}_N^{Ort} F(\theta)$ is sub-Gaussian with parameter $\frac{1}{\sqrt{T}} \sqrt{\frac{\beta^2 c^2 R^2}{d^2 \sigma^2} + \frac{R^2 B^2}{4\sigma^2 d}} = \frac{1}{\sqrt{N}} \sqrt{\frac{\beta^2 c^2 R^2}{d\sigma^2} + \frac{R^2 B^2}{4\sigma^2}}$ .

And therefore:

**Corollary 11.10.** The orthogonal gradient estimator $\hat{\nabla}_N^{Ort} F(\theta)$ satisfies the following concentration inequality:

$$\mathbb{P}\left( \max_{j=1,\cdots d} \left| \left( \hat{\nabla}_N^{Ort} F(\theta) \right)_j - \left( \mathbb{E}\left[ \hat{\nabla}_N^{Ort} F(\theta) \right] \right)_j \right| \geq t \right) \leq 2de^{\frac{-t^2 N \sigma^2}{\frac{\beta^2 c^2 R^2}{d} + \frac{R^2 B^2}{4}}} .$$

This argument finalizes the proof of Theorem 4.3. Whenever the lengths of the scalings $B, \beta_i$ are of order $O(1)$, we recover a concentration rate of order $O(d\exp(-t^2 N))$, which is comparable to the rate for the vanilla estimator. The analysis of orthogonal estimators is substantially harder than in the vanilla case due to the non i.i.d nature of the sampling process. This is to our knowledge the first result of its type.

## 12 Experiments: further details for variational autoencoder implementation

In this section, we give further details on the setup of the variational autoencoder experiments on MNIST appearing in Section 5.2 of the main paper.

### 12.1 Architectures

We use a 64-dimensional $N(0, I)$ distribution for the prior over the latent state $z$. The generative model $p_\theta(x|z)$ is specified by a fully-connected neural network with 64 input units and two hidden layers of 500 units. The hidden unit activation functions are ReLUs, and the final layer activations are sigmoids. A Bernoulli likelihood is used to train the output of the network. Here, $\theta$ represents the trainable parameters of the network. The recognition model $q_\phi(z|x)$ is also given by a fully connected neural network with two hidden layers of 500 units. The hidden layer activation functions are ReLUs, and the final layer is linear, outputting a mean vector $\mu_\phi(x)$ and a log-standard deviation vector $\log(\sigma_\phi(x))$, which parametrise an approximate factorised Gaussian posterior $N(\mu_\phi(x), \sigma_\phi^2(x))$ for the latent encoding given $x$. Here, $\phi$ are the trainable parameters of the network. We initialise all weights of the networks using the normalised initialisation of (Glorot and Bengio, 2010), and initialise biases to 0.

### 12.2 Results

Table 1: Train and test ELBO achieved with various sampling algorithms. iid refers to independently sampled directions, in contrast to ort, which refers to orthogonally constrained directions as in Algorithm 2. anti-eq corresponds to antithetic pairs of samples with matching norms (as in Algorithm 2, and anti-inv corresponds to antithetic pairs with norm couplings as in Algorithm 1.

| | $m = 2$ | | | | | |
|---|---|---|---|---|---|---|
| | iid | iid-anti-eq | iid-anti-inv | ort | ort-anti-eq | ort-anti-inv |
| Train | -99.59 | -98.60 | -98.64 | -99.30 | -98.56 | **-98.47** |
| Test | -99.96 | -99.02 | -99.05 | -99.69 | -98.97 | **-98.94** |

| | $m = 8$ | | | | | |
|---|---|---|---|---|---|---|
| | iid | iid-anti-eq | iid-anti-inv | ort | ort-anti-eq | ort-anti-inv |
| Train | -98.88 | -98.48 | -98.55 | -98.79 | **-98.41** | -98.50 |
| Test | -99.26 | -98.92 | -98.98 | -99.24 | **-98.89** | -98.95 |

We train on minibatches of 50 images, and use the Adam optimiser with a learning rate of $10^{-4}$, and all other parameters set to default settings; the learning rate was softly optimized for the performance of the i.i.d. method. It is thus possible that with further hyperparameter tuning for each individual sampling method, further improvements in performance may be observed for GCMC sampling schemes; intuitively, we might expect that the variance reduction in stochastic gradients that GCMC methods achieve would allow a larger learning rate to be used. We report average test and train log-likelihood after 50 epochs of training, to assess the impact of the considered sampling schemes on the speed of learning for the model. We summarize full results for a variety of sampling methods in Table 1. We note that GCMC methods always improve speed of training relative to i.i.d., and in general the most substantial improvement combines some variant of antithetic sampling with orthogonality constraints.

# 13    Experiments: Learning efficient navigation policies with ES strategies

In this section we give additional information on the ES gradient estimators described in Section 5.1,as well as a description of the video library that we attach to the paper, and additional experimental results.

## 13.1    Estimator specification

The vanilla ES gradient estimator is given by

$$\hat{\nabla}_N^V F_\sigma(\theta) = \frac{1}{N\sigma} \sum_{i=1}^{N} F(\theta + \sigma\epsilon_i)\epsilon_i, \text{ where } \epsilon_i \sim \mathcal{N}(0, I) \text{ are all i.i.d.} .$$

We consider three variants of control variates: forward finite-difference, in which the estimator is given by

$$\frac{1}{N\sigma} \sum_{i=1}^{N} (F(\theta + \sigma\epsilon_i) - F(\theta))\epsilon_i \,,$$

antithetic, in which the estimator is given by

$$\frac{1}{2N\sigma} \sum_{i=1}^{N} (F(\theta + \sigma\epsilon_i) - F(\theta - \sigma\epsilon_i))\epsilon_i \,,$$

and antithetic-coupled, in which the estimator is given by

$$\frac{1}{2N\sigma} \sum_{i=1}^{N} (F(\theta + \sigma\epsilon_i)\epsilon_i + F(\theta + \sigma\epsilon_i')\epsilon_i' - F(\theta)(\epsilon_i + \epsilon_i')) \,,$$

where $\varepsilon_i$ and $\varepsilon_i'$ are coupled as in Algorithm 1. Note the additional term dependent on $F(\theta)$ appearing in the antithetic $-$ coupled estimator, in order to cancel the zeroth order term in the Taylor expansion of the above objective.

## 13.2    Video library

We attach to the paper a collection of videos showing how policies learned with different tested MC algorithms work in a simulator. Each file is in a .webm format and its name is using the following template: sensing_mechanism-control_variate-samples_number, where: sensing_mechanism stands for the sampling strategy and is chosen from the set: {MCGaussian, MCGaussianOrthogonal, MCRandomHadamard}, control_variate-samples_number stands for the type of the control variate term used and is chosen from the set: {vanilla, forward finite-difference, antithetic and antithetic-coupled} and finally: samples_number stands for the number of samples used in the MC algorithm at each iteration of the optimization routine to approximate current gradient vector. These videos serve to illustrate the types of policies learnt under the variety of sampling mechanisms considered in Section 5.1.

## 13.3 RL experiments: additional results

We present here the results of all experiments conducted to learn good quality navigation policies for the Minitaur platform for the following sampling strategies: MCGaussian, MCGaussianOrthogonal, MCRandomHadamard, the following control variate terms: vanilla, forward finite-difference, antithetic and the following number of samples: $m = 8, 48, 96$.

(a) MCGaussian-antithetic-48

(b) MCGaussian-antithetic-96

(c) MCGaussian-forward_fd-8

(d) MCGaussian-forward_fd-48

(e) MCGaussian-forward_fd-96

(f) MCGaussian-vanilla-8

Figure 3: Additional experimental results showing training curves for different MC algorithms. Naming is borrowed from the video library section.

(a) MCGaussian-vanilla-48

(b) MCGaussian-vanilla-96

(c) MCGaussianOrthogonal-antithetic-16

(d) MCGaussianOrthogonal-antithetic-32

(e) MCGaussianOrthogonal-antithetic-48

(f) MCGaussianOrthogonal-antithetic-64

Figure 4: Additional experimental results showing training curves for different MC algorithms. Naming is borrowed from the video library section.

(a) MCGaussianOrthogonal-antithetic-96

(b) MCGaussianOrthogonal-forward_fd-8

(c) MCGaussianOrthogonal-forward_fd-48

(d) MCGaussianOrthogonal-forward_fd-96

(e) MCGaussianOrthogonal-vanilla-8

(f) MCGaussianOrthogonal-vanilla-48

Figure 5: Additional experimental results showing training curves for different MC algorithms. Naming is borrowed from the video library section.

(a) MCGaussianOrthogonal-vanilla-96

(b) MCRandomHadamard-vanilla-8

(c) MCRandomHadamard-vanilla-48

(d) MCRandomHadamard-vanilla-96

(e) MCRandomHadamard-forward_fd-8

(f) MCRandomHadamard-forward_fd-48

Figure 6: Additional experimental results showing training curves for different MC algorithms. Naming is borrowed from the video library section.

(a) MCRandomHadamard-forward_fd-96

(b) MCRandomHadamard-antithetic-8

(c) MCRandomHadamard-antithetic-48

(d) MCRandomHadamard-antithetic-96

Figure 7: Additional experimental results showing training curves for different MC algorithms. Naming is borrowed from the video library section.