[Reviews · NeurIPS 2018]

Reviewer 1



Summary: -------------- This paper proposes a number of expectation estimation strategies which strive to attain lower error than naive methods relying on iid sampling. This is especially important in settings where evaluating the objective function at the sampled variates is expensive (e.g. RL problems.) The advantage of Monte Carlo over deterministic methods is that theoretical assurances are more readily obtained. The approach is based on (numerically) finding optimal couplings, i.e. joint distributions over an augmented state space marginalizing to the expectation distribution of interest. While a uniformly optimal coupling does not exist over generic function classes, as is appreciated in statistical decision theory, the problem is well-defined in both expected and minimax variants. The optimizer to the K-expected loss is shown to minimize a multi-marginal transport problem, with the nice result that the solutions are space-filling, bringing to mind the behavior of QMC methods. In Proposition 2.7, a relaxation of the minimax problem is obtained when F is the unit ball in a RKHS associated with K, which interestingly turns out to solve the expected loss under the GP prior with covariance K. Algorithm 2 shows how to optimally couple m antithetic pairs in accordance with Theorem 2.10, which states that sequential orthogonal sampling achieves this under an isotropic marginal. Section 3 connects geometrically-coupled sampling with low-discrepancy sequences, and Section 4 provides some instructive concentration bounds for both vanilla ES gradient estimators and orthogonal estimators. This paper provides a number of valuable theoretical contributions with important practical implications, and I recommend its acceptance to NIPS. While the topic is not my field of research, the ideas in the paper were in my view sufficiently well-explained and motivated for me to appreciate the problems and solution. Issues/Questions: ----------------- The intended domain of application is when evaluation time of f dominates the process. While Algorithm 1 is clearly efficient, is it possible to give a sense of the cost generating conditionally orthogonal samples as in Algorithm 2? Clarity: -------------- The paper is very well-written and clear;. Significance: ------------ In addition to its algorithmic contributions, this paper provides several explanations of the success of currently used approximation methodologies. The restriction of the marginal to be isotropic is strong and prohibits applicability in other contexts, however the case is convincingly made that in ES-based policy learning and VAE ELBO optimization, the method is practically relevant. Technical Correctness: --------------------- I did not painstakingly verify each detail of the proofs, but the results are technically correct as far as I could tell. Minor: ------- - Why does "s" refer to number of samples in the experiments while "m" in the rest of the paper? - l186: sampes -> samples

Reviewer 2



The paper proposes a new approach to drawing a sequence of samples for Monte Carlo estimation. The method is motivated from optimal transport theory and the authors develop theoretical properties of the sampling algorithm in great detail. An interesting results are established in relation to quasi Monte Carlo samples as well as for evolution strategies gradient estimator. Frankly, I am not very familiar with optimal transport theory so I approached the paper as a student would in order to make sense of section 2. It took me some time to parse it all but at the end of the day, I think the theory is presented in a logical fashion. However, I have not yet fully understood the implication of theorem 2.10. I think this theorem is the one that gave rise to algorithm 2, which aims to generate orthogonal to samples (from $\eta$). However, it does not seem like the samples generated according to Algorithm 2 are guaranteed to be generated from $\mu^*$ (where $\mu^*$ is a solution to $K$-optimal coupling) because Theorem 2.10 states that if you have samples from $\mu^*$, then it satisfies the orthogonality. It would be helpful to provide a bit on the connection between Algorithm 2 and Theorem 2.10. Section 3 establishes result for discrepancy of the samples generated. Seciont 4 establishes concentration inequalities for the evolution strategy gradient estimator, which has practical implications in reinforcement learning problems. The experiments are carried are interesting. Minor comments: Line 131: Definition 3 -> Definition 2.2 or Equation (3)? More explanation behind the video library experiments in the Appendix would be helpful. It was not obvious what I should be looking for in these videos.

Reviewer 3



This article considers a coupling approach to MC estimation, based upon the so-called K-optimality criterion to minimise the MSE over a class generated by a GP of kernel K. This is in slightly flawed from the beginning because obviously one doesn't want to sacrifice reductions in MSE for massive increases in costs. However, accepting this is a sensible way to go forward, a variety of possibly interesting theorems are presented about the idea and simulations given. Basically, this paper could be really quite interesting, if it were a long journal article where each aspect is clearly explained and considered. As it stands, to me, it feels like an unfocussed mess.